# Potential-dependent transition of reaction mechanisms for oxygen evolution on layered double hydroxides

Zeyu Wang [1], William A. Goddard III[2] & Hai Xiao [1] ✉

Oxygen evolution reaction (OER) is of crucial importance to sustainable energy and environmental engineering, and layered double hydroxides (LDHs) are among the most active catalysts for OER in alkaline conditions, but the reaction mechanism for OER on LDHs remains controversial. Distinctive types of reaction mechanisms have been proposed for the O-O coupling in OER, yet they compose a coupled reaction network with competing kinetics dependent on applied potentials. Herein, we combine grand-canonical methods and micro-kinetic modeling to unravel that the nature of dominant mechanism for OER on LDHs transitions among distinctive types as a function of applied potential, and this arises from the interplay among applied potential and competing kinetics in the coupled reaction network. The theory-predicted overpotentials, Tafel slopes, and findings are in agreement with the observations of experiments including isotope labelling. Thus, we establish a computational methodology to identify and elucidate the potential-dependent mechanisms for electrochemical reactions.

Oxygen evolution reaction (OER) is a universal anodic half-reaction that oxidizes $H_2O$ to $O_2$, and it is widely employed to couple with cathodic half-reactions that are of great values to energy and environmental engineering for a sustainable future, including the hydrogen evolution reaction and $CO_2$ reduction reaction that compose the water splitting[1–5] and artificial photosynthesis[6–8], respectively. Although OER is a key component in the applications of electrochemistry in energy and environment, its sluggish kinetics results in a major efficiency loss[4,5,9–11], which hinders its applications and thus requires developing highly efficient electrocatalysts[11–13]. Consequently, great efforts have been made to formulate a fundamental understanding of OER mechanisms that is key to the rational design of high-performing OER electrocatalysts for practical applications[14–22].

Layered double hydroxides (LDHs) based on $3d$ transition metals (TMs) including Fe, Co, and Ni are among the most efficient OER electrocatalysts in alkaline environments[23–25]. There has been an extensive literature on developing efficient and robust $3d$-TM-based LDHs for catalyzing the alkaline OER, among which the Fe-doped Ni-based LDHs ($Ni(Fe)O_xH_y$) are reported as the most active with a Fe/Ni

ratio of $1/3$[24,26–29], and the OER activities of Ni-based LDHs are with the order of $NiO_xH_y \ll Ni(Co)O_xH_y < Ni(Fe)O_xH_y$[11].

Despite the state-of-the-art OER performance, a key question regarding the Ni-based LDH catalysts remains, i.e., the OER mechanisms are not elucidated. For the key elementary step of O–O coupling in the OER mechanisms, there are three distinctive types of reaction channels[11,12,30] that have been proposed, including the adsorbate evolution mechanism (AEM)[24,31], the intramolecular oxygen coupling (IMOC)[32–34], and the lattice oxygen mechanism (LOM)[15,35–37]. Experimental techniques such as isotope labeling can give hints on the oxygen source of $O_2$ in order to determine the mechanism, but the elusive nature of intermediates renders the experimental evidences inconclusive. Moreover, OER involves the oxidation of water molecule into $O_2$ that requires a large electrochemical driving force, and thus it is intriguing to understand how the O–O coupling depends on the applied potential.

Theoretical modeling can provide an atomistic understanding and thus play an essential role in elucidating the reaction mechanisms for OER[24,36,38–44]. However, the three distinctive types of pathways, i.e., the

[1]Department of Chemistry and Key Laboratory of Organic Optoelectronics and Molecular Engineering of Ministry of Education, Tsinghua University, Beijing 100084, China. [2]Materials and Process Simulation Center, California Institute of Technology, Pasadena, CA 91125, USA. ✉e-mail: haixiao@tsinghua.edu.cn

AEM, IMOC, and LOM, are actually three idealizations of a single complex reaction network with competing kinetics among its coupled sub-pathways and it is further complicated by the dependence of kinetics on the applied potentials[32]. There have been previous theoretical studies of OER[45,46] that compared kinetics between AEM and LOM in a decoupled scenario. But there have been no theoretical studies so far that present an understanding of this whole coupled reaction network on Ni-based LDHs with explicit considerations of kinetics and applied potentials, which is key to unraveling the true mechanisms underlying the catalytic performance for OER by Ni-based LDHs.

In this work, we combine grand-canonical methods and micro-kinetic modeling to investigate the interplay among the competing kinetics and applied potentials in the coupled reaction network composed of AEM, IMOC, and LOM types of mechanisms for OER on M-doped Ni-based LDHs (Ni(M)OOH, M = Ni, Co, and Fe). We unravel that the nature of dominant mechanism for OER on Ni(M)OOH transitions among distinctive types as a function of both the catalyst composition and the applied potential, which arises exactly from the interplay among the applied potential and competing kinetics in the coupled reaction network. The predicted overpotentials and Tafel slopes are in excellent agreement with the experimental values, and

our results are also consistent with the observations of isotope-labeling experiment. Thus, by recognizing the coupled reaction network and investigating its potential-dependent kinetics, we establish a computational methodology to make accurate predictions and elucidate the mechanisms for OER and other electrochemical reactions with potential-dependent mechanisms. In addition, we identify the spin densities on both the metal site and the reactive surface O species to be delicate descriptors of the OER activity on LDHs.

## Results

### Formulating the reaction network for OER on Ni(M)OOH

We consider all the three distinctive types of OER mechanisms on Ni(M)OOH under alkaline conditions, i.e., AEM, IMOC, and LOM, each of which is composed of four electrochemical steps and one O−O coupling chemical step (Fig. 1a−c). Note that the skeletal formulas of active sites in Fig. 1 illustrate only the key dual-metal centers, which contain two neighboring TM sites connected by two lattice O atoms ($O^{latt}$), and more details on the models can be found in Supplementary Information (SI).

The AEM has been widely studied via considering only the electrochemical steps with a thermodynamics-only scheme[47]. Based on a

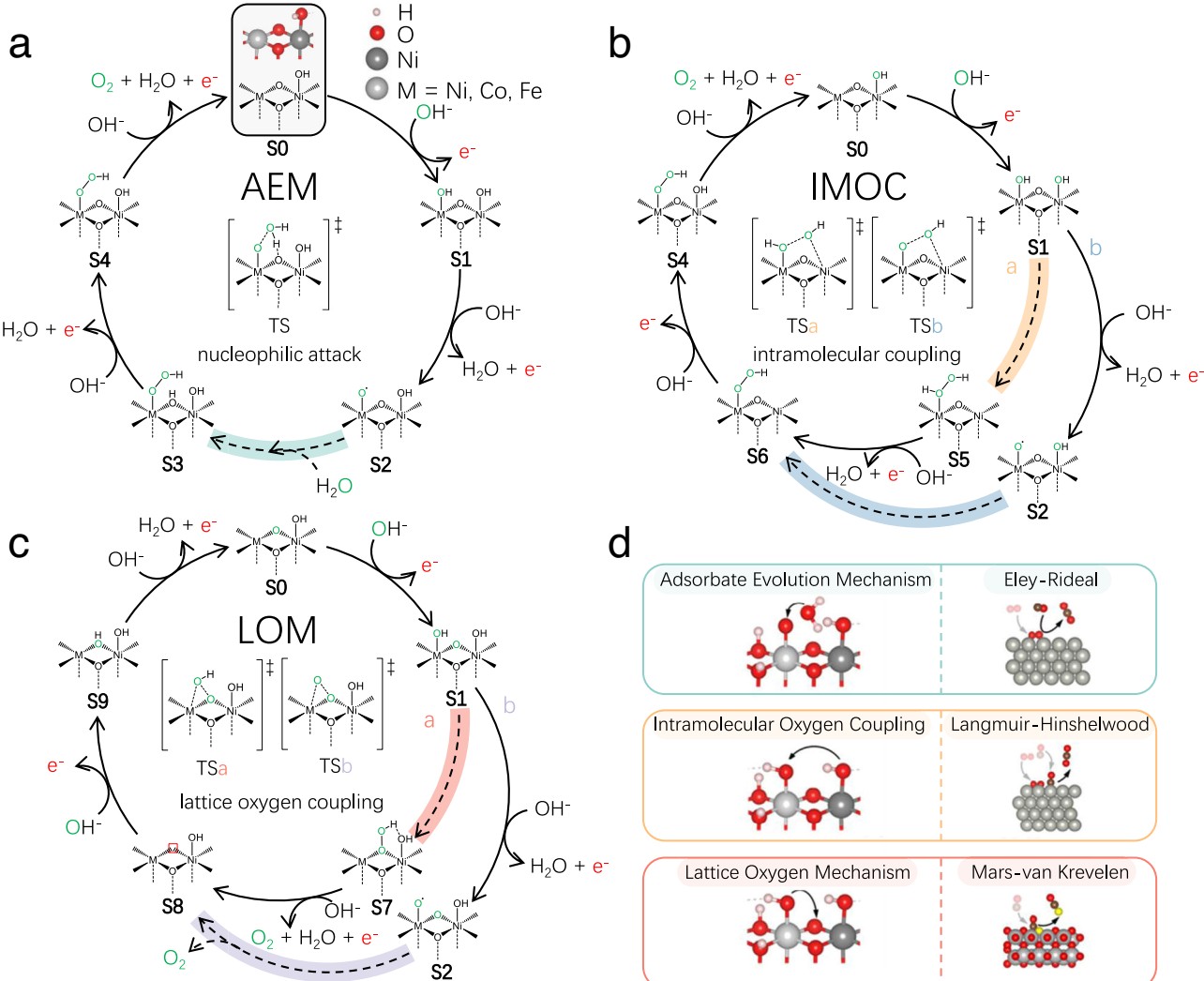

**Fig. 1 | OER mechanisms.** Distinctive types of reaction mechanisms for OER on Ni(M)OOH under alkaline conditions, including **a** AEM, **b** IMOC, and **c** LOM. All states are labeled by **S$n$**, and the O−O coupling chemical steps are represented by dashed lines and highlighted. O atoms marked in green are the source of $O_2$ product in each cyclic route. The TS of O−O coupling step is shown in the middle of

each cyclic route to feature the key differences among different types of mechanisms. Note that the O−O coupling steps in (**a**−**c**) resemble the three typical mechanisms in heterogeneous catalysis, i.e., the Eley-Rideal (ER), Langmuir-Hinshelwood (LH), and Mars-van Krevelen (MvK) mechanisms, as shown in (**d**).

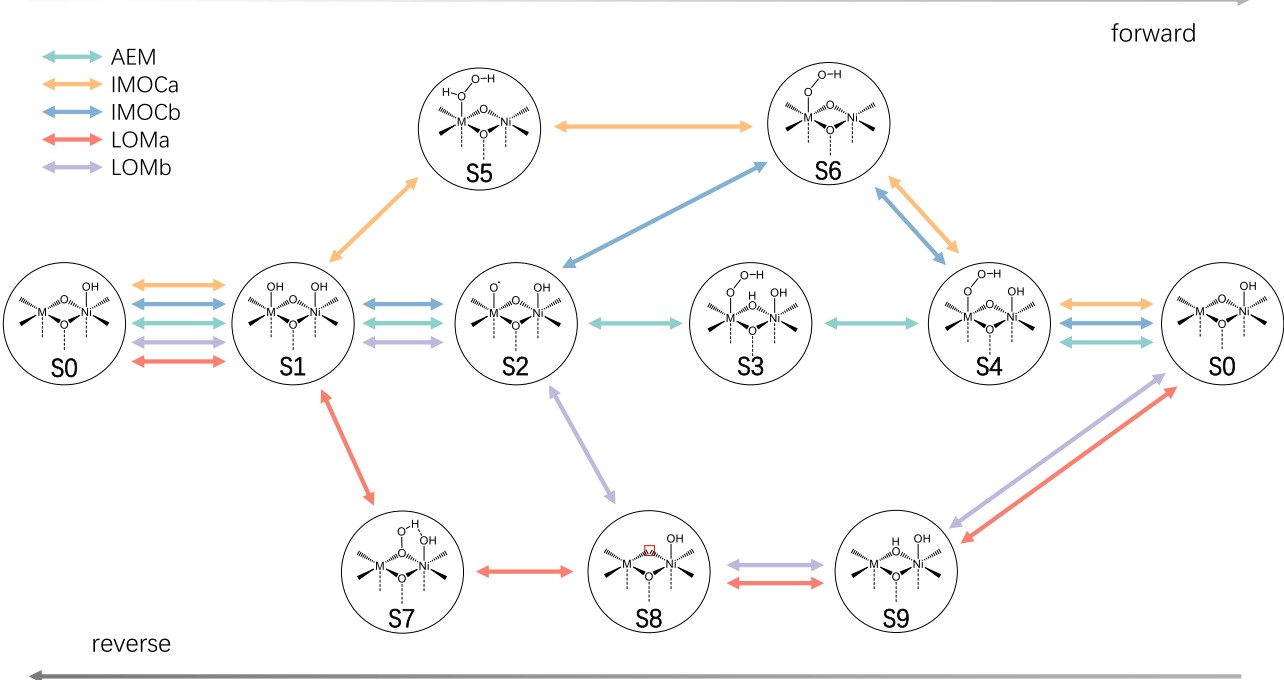

**Fig. 2 | Reaction network for OER.** The coupled reaction network formed by all the reaction mechanisms for OER on Ni(M)OOH. Note that both the forward and reverse reactions are fed to the microkinetic modeling.

more recent study[38], we formulate the AEM route on Ni(M)OOH (Fig. 1a), which includes the kinetics of O–O coupling chemical step that is enabled by the generated adsorbed O radical (*O•, **S2** in Fig. 1a). The AEM here starts with electrochemical oxidation of OH⁻ to the adsorbed OH (*OH, **S1**), followed by electrochemical deprotonation of *OH to form *O• (**S2**). Note that the radical nature of *O• is further confirmed by spin population analysis (Figure S1), consistent with previous experiments[38,48,49], and it was suggested to reduce the O–O coupling barrier[50]. This highly active *O• and the neighboring $O^{latt}$ synergistically attack a $H_2O$ molecule in the electrolyte to form *OOH and $O^{latt}H$ (**S3**), thus delivering the O–O coupling chemical step. The two following electrochemical deprotonation steps recover $O^{latt}$ (**S4**) from $O^{latt}H$ and produce $O_2$ from *OOH sequentially.

Alternatively, the IMOC route on Ni(M)OOH (Fig. 1b) couples the neighboring adsorbates directly to deliver the O–O coupling, and there are two branches of the IMOC route. In one branch (labeled as a), the O–O coupling takes place between two neighboring *OH in **S1**, while in the other branch (labeled as b), **S2** with *O• is first generated and then the O–O bond is formed between *O• and its neighboring *OH.

The LOM route, in contrast to the above two, couples the bridging $O^{latt}$ of catalyst surfaces with the neighboring adsorbate to deliver the O–O coupling, and there are also two branches of the LOM route. In one branch (labeled as a), $O^{latt}$ couples with *OH in S1, while in the other branch (labeled as b), $O^{latt}$ couples with *O• after **S2** is formed. The consumed $O^{latt}$ is recovered sequentially by electrochemical oxidation and deprotonation of OH⁻.

It is worth noting that the key O–O coupling steps in the three types of OER mechanisms can be well categorized by the three typical mechanisms in heterogeneous catalysis, i.e., the Eley-Rideal (ER), Langmuir-Hinshelwood (LH), and Mars-van Krevelen (MvK) mechanisms (Fig. 1d). Specifically, the O–O coupling in AEM involves the adsorbed *O• to react with a $H_2O$ molecule in the electrolyte, resembling the ER mechanism; the O–O coupling in IMOC involves only the adsorbed reactants, which is characteristic for the LH mechanism; the O–O coupling in LOM consumes the $O^{latt}$ of catalyst, and this exactly belongs to the MvK mechanism. Thus, the three types of OER

mechanisms compose a comprehensive set for understanding the OER on LDHs.

We note that this analogy is applicable to other electrochemical reactions such as the hydrogen evolution reaction: the Volmer-Heyrovsky and Volmer-Tafel mechanisms resemble closely the ER and LH mechanisms, respectively, and the MvK mechanism is a possible channel on catalysts that can form bulk hydrides, such as Pd[51,52]. This analogy to the typical types of mechanisms in heterogeneous catalysis may serve as a useful scheme to categorize and identify mechanisms for electrochemical reactions.

**Predicting the OER activity on Ni(M)OOH**

All the OER mechanisms in Fig. 1 are interconnected and thus form a coupled reaction network (Fig. 2), in which the competition among different routes is enabled by their common intermediates. Thus, we investigate the coupled reaction network with microkinetic modeling that includes competing kinetics of all reaction pathways and their dependence on the applied potentials, as described in Methods and SI. Note that we investigate explicitly the reaction kinetics, i.e., the transition state (TS) and associated energy barrier, of the O–O coupling chemical step.

We first investigate the free energy profiles of the mechanisms on each Ni(M)OOH LDH, and the results are shown in Fig. 3a–c for M = Ni, Co, and Fe, respectively. Note that the reaction free energies (Ω) and barriers (Ω) are all presented with values at the equilibrium potential for OER, i.e., $U = 1.23$ V (RHE scale with pH = 14) in the following discussions for simplicity, and their explicit dependences of $U$ are listed in Table S3. The potential limiting step (PLS) is the electrochemical step that possesses the maximal Ω in each pathway, and $Ω_{PLS}$ has been widely adopted to calculate the overpotential ($η$) approximately and evaluate the efficiency of electrochemical reaction (the PLS scheme)[34]. Thus, we consider both the PLS scheme (Figure S2) and the results of our microkinetic modeling (Figure S3 and Fig. 3d). Note that under the PLS scheme, AEM and IMOCb cannot be distinguished.

On NiOOH, the LOMa pathway renders the lowest $Ω_{PLS}$ of 0.48 eV under the PLS scheme (Figure S2a), with the PLS from **S9** to **S0** that

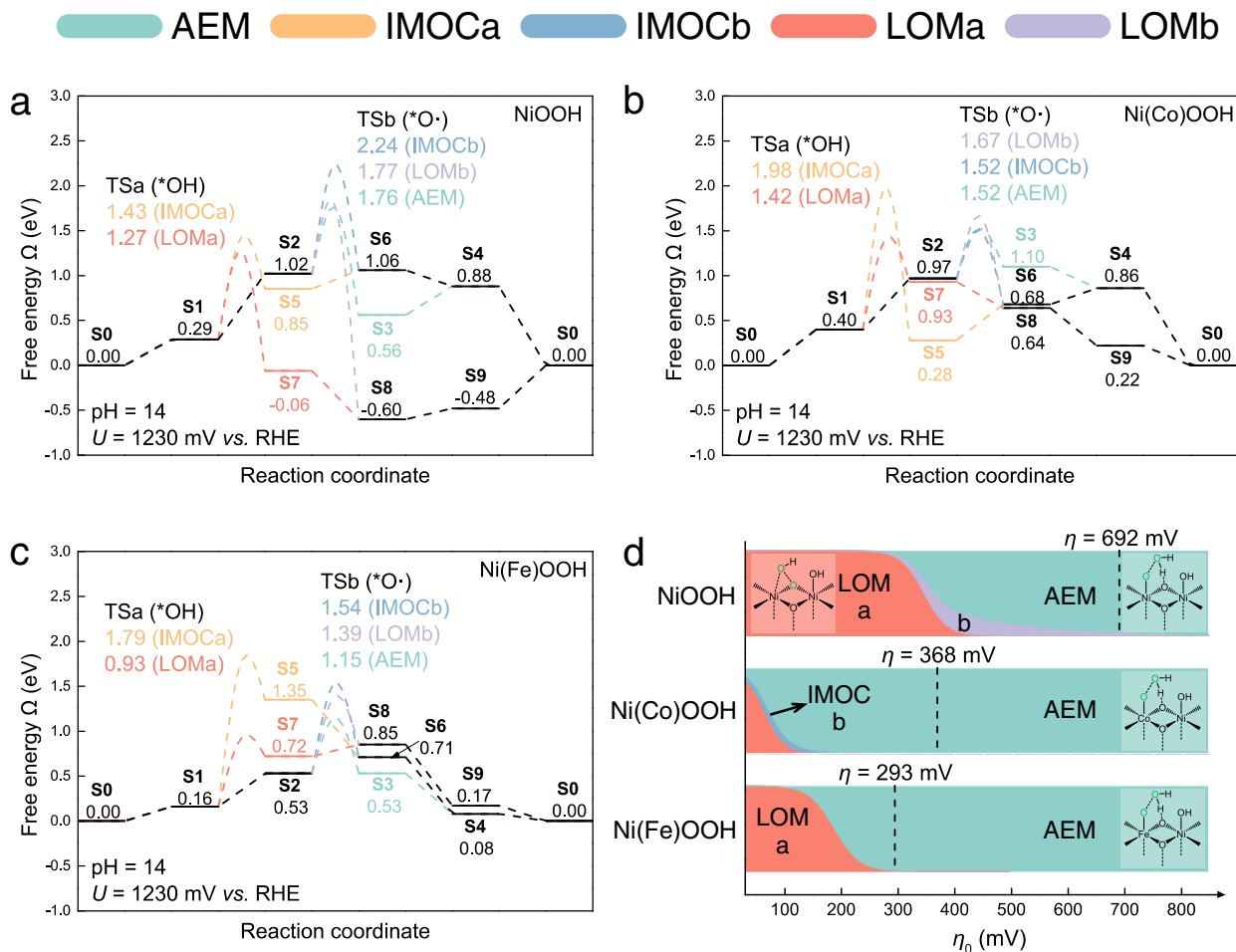

**Fig. 3 | OER free energy profiles and microkinetic modeling results.** Grand free energy profiles at the equilibrium potential for OER on **a** NiOOH, **b** Ni(Co)OOH, and **c** Ni(Fe)OOH following different reaction mechanisms. **d** The dominant mechanism for OER on Ni-based LDHs as a function of both the catalyst composition and the applied potential ($\eta_0 = (U - 1230)$ mV) predicted by our microkinetic modeling scheme. The proportions of colors in the intersection at each $\eta_0$ correspond to the proportions of contributions from different types of mechanisms.

exposes the bridging $O^{latt}$ from deprotonating $O^{latt}H$, and thus the PLS scheme predicts LOMa to be the dominant mechanism underling OER on NiOOH. However, our microkinetic modeling results (Fig. 3a and Figure S3a, b) show that, although it is indeed the LOMa pathway that dominates the contribution to $j$ at low applied potentials ($\eta_0$, defined as $(U - 1230)$ mV for convenience to locate $\eta$), it is overtaken by the AEM pathway for $\eta_0 > 365$ mV, and it is the AEM pathway that determines the $\eta$ of 692 mV at $j = 10$ mA·cm⁻². This demonstrates explicitly that the nature of dominant mechanism underlying the catalytic performance of OER on Ni-based LDHs is strongly coupled with the applied potential and subject to change.

The predominance of LOMa for OER on NiOOH at low $\eta_0$ arises from the lowest $\Omega_{PLS}$ as well as a viable $\Omega$ of 0.98 eV for its O−O coupling chemical step from **S1** to **S7** that couples *OH with $O^{latt}$ (Fig. 3a). This enables a feasible kinetics for OER to proceed through the LOMa pathway to consume **S1**, even though its PLS remains endothermic (requiring 480 mV for its $\Omega_{PLS}$ of 0.48 eV) at low $\eta_0$.

The AEM, IMOCb, and LOMb pathways on NiOOH all render the same $\Omega_{PLS}$ of 0.73 eV under the PLS scheme (Figure S2a), with a shared PLS from **S1** to **S2**, and thus the PLS scheme predicts that all the three pathways contribute equally to deliver the OER, but much less than the LOMa pathway. As $\eta_0$ increases, their PLS is turned on, which forms the active *O• that can drive the O−O coupling in AEM, IMOCb, and LOMb (Fig. 3a) with $\Omega$ of 0.74, 1.22, and 0.75 eV, respectively. Thus, *O• enables fast kinetics of O−O coupling in both the AEM and LOMb

pathways. This drains the concentration of **S1** exponentially, converting **S1** to **S2** for driving the O−O coupling in AEM and LOMb, which leads to the sharp decay in the contribution of LOMa that requires **S1** and the fast rise in the contributions of AEM and LOMb to $j$ (Figure S3a and its inset), resulting in the transition of dominant mechanism.

However, as $\eta_0$ increases further, the contributions of AEM and LOMb are sharply differentiated, and eventually, AEM becomes the dominated mechanism for OER (Figure S3a, b). This arises from the different potential-dependences of $\Omega$ for O−O coupling in the two pathways, as listed in Table S3. The $\Omega$ in AEM decreases with $\eta_0$ faster (a more negative coefficient of $\eta_0$) than that in LOMb. In contrast, IMOCb makes little contribution to the OER activity, due to a large $\Omega$ of 1.22 eV for O−O coupling with a positive coefficient of $\eta_0$ (Table S3).

Therefore, the interplay among the applied potential and the competing kinetics within the coupled reaction network plays the key role in the OER on Ni-based LDHs, and it is exactly this interplay that results in the transition of dominant mechanism for OER on NiOOH LDH from LOM to AEM.

On Ni(Co)OOH, the IMOCa and LOMa pathways render the lowest $\Omega_{PLS}$ of 0.40 eV (Figure S2b), sharing the same PLS from **S0** to **S1**. Similar to OER on NiOOH, the AEM, IMOCb, and LOMb pathways on Ni(Co)OOH share a higher $\Omega_{PLS}$ of 0.57 eV than LOMa with the PLS from **S1** to **S2**, which requires deeper oxidation. As $\eta_0$ increases over 70 mV, even though the PLS is far from turned on ($\Omega_{PLS}$ of 0.57 eV requires an $\eta_0$ of 570 mV), the increase in the concentration of **S2** with the active

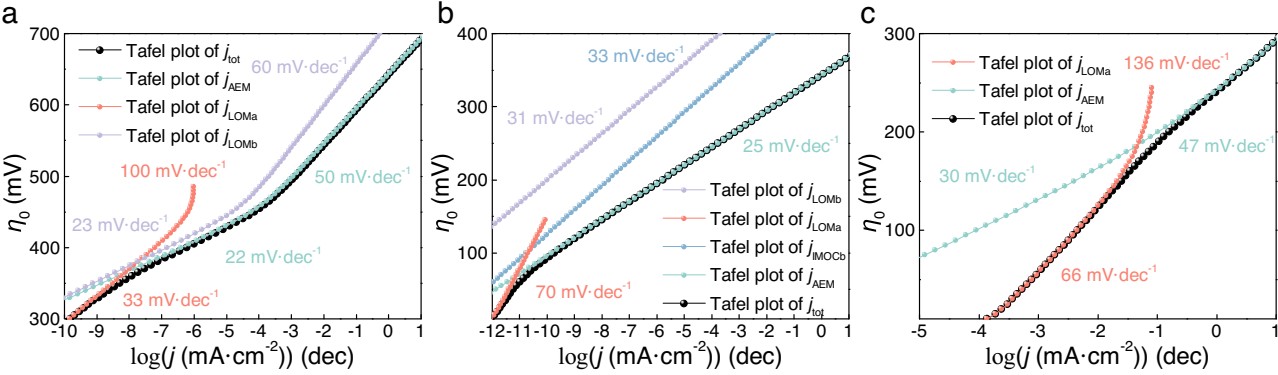

**Fig. 4 | Calculated Tafel plots for OER.** Calculated Tafel plots and slopes of **a** NiOOH, **b** Ni(Co)OOH, and **c** Ni(Fe)OOH, with components by different types of mechanisms in the coupled reaction network. Note that certain component Tafel plots are off the chart and thus not shown.

*O•, although remaining negligibly small (Figure S4b), is yet able to drive the AEM, IMOCb, and LOMb pathways, owing to their favorable $\Omega$ of 0.56, 0.55, and 0.70 eV, respectively (Fig. 3b). Thus, after $\eta_0 > 70$ mV, the dominant mechanism changes from LOMa to AEM (Fig. 3d) which possesses a much lower barrier for the O–O coupling to deliver the OER by consuming *O•.

Sharing the active *O• for O–O coupling with the lowest barrier, the IMOC pathway is yet gradually suppressed by the AEM pathway that contributes an increasingly significant portion to $j$ (Figure S3c, d), which arises from the fact that the $\Omega$ in AEM has a much more negative coefficient of $\eta_0$ than those in IMOCb and LOMb (Table S3). Note that the LOMb pathway starts to make increasingly significant contribution at high $\eta_0$ (Figure S3c), although it remains at a vanishingly small percentage (Figure S3d). This is because **S9** in the LOM pathways has the most negative coefficient of $\eta_0$ on Ni(Co)OOH (Table S3), gaining a beneficial thermodynamic equilibrium for LOMb as $\eta_0$ increases, but the $\Omega$ and thus the kinetics of LOMb cannot compete with that of AEM.

On Ni(Fe)OOH, which is the best OER catalyst among Ni(M)OOH LDHs, the AEM, IMOCb, and LOMb pathways render the lowest $\Omega_{PLS}$ of only 0.37 eV under the PLS scheme (Figure S2c), implying a facile PLS from **S1** to **S2** that forms *O•. While with including the O–O coupling chemical step enabled by *O•, AEM, IMOCb, and LOMb (Fig. 3c) present drastically different $\Omega$ of 0.62, 1.02, and 0.87 eV, respectively, and this indicates that their kinetics follow a clear order of AEM > LOMb > IMOCb. Indeed, our microkinetic modeling results (Figure S3e, f and Fig. 3d) show that the AEM dominates the contribution to $j$ for $\eta_0 > 185$ mV, with others remaining marginal at all $\eta_0$, and this simply arises from the fact that AEM has the lowest $\Omega$ among all mechanisms.

Interestingly, although LOMa on Ni(Fe)OOH renders an unfavorable $\Omega_{PLS}$ of 0.69 eV, our microkinetic modeling shows a transition of dominant mechanism with LOMa dominating the contribution to $j$ at low $\eta_0$ (Fig. 3d). This arises from the introduction of O–O coupling step in LOMa that divides the PLS into a chemical step and an electrochemical step (Fig. 3c), effectively reducing the original $\Omega_{PLS}$ to only 0.16 eV (with the PLS from **S0** to **S1**) and a viable $\Omega$ of 0.77 eV. Thus, at low $\eta_0$, LOMa outcompetes AEM to deliver the OER due to a lower $\Omega_{PLS}$, but AEM takes over soon with a faster kinetics as $\eta_0$ increases.

Although there have been previous theoretical studies[45,46] to compare kinetics between AEM and LOM in a decoupled scenario and conclude that the nature of dominant mechanism is composition-dependent, our computational methodology explicitly investigates the competing and potential-dependent kinetics in the coupled reaction network composed of AEM, IMOC, and LOM, which leads to the finding that the nature of dominant mechanism is also dependent of the applied potential. Figure 3d illustrates our finding that the nature of dominant mechanism for OER on Ni-based LDHs transitions among distinctive types as a function of both the catalyst composition and the

applied potential, and predicts the $\eta$ of 692, 368, and 293 mV at $j = 10$ mA·cm$^{-2}$ for OER on Ni(M)OOH with M = Ni, Co, Fe, respectively.

It seems universal that the dominant mechanism for OER on all Ni(M)OOH transitions from LOM to AEM as the applied potential increases, with thin contributions from LOMb and IMOCb at the transitional regions on NiOOH and Ni(Co)OOH, respectively. This arises from the general fact that LOM (LOMa in particular) is first turned on owing to the lowest $\Omega_{PLS}$ and a viable barrier for O–O coupling at low $\eta_0$, but as $\eta_0$ increases, AEM takes over eventually owing to its fastest kinetics with the lowest barrier and its negative coefficient of $\eta_0$ for O–O coupling. This drains the concentration of **S1** exponentially, which leads to the sharp decay in the contribution of LOMa that requires **S1** and the fast rise in the contributions of AEM with converting **S1** to **S2** for driving the O–O coupling. Therefore, the interplay among the applied potential and the competing kinetics within the coupled reaction network plays the key role in delivering the transition of reaction mechanisms for OER on LDHs.

It has been well established that the Tafel slope is directly connected to the electrochemical reaction mechanism, and a transition in the type of reaction mechanism may lead to the variation in the Tafel slope[53–55]. However, there are subtleties in the significance of the variation in the Tafel slope, which may arise from a few different types of changes, including the change in the rate limiting step within the same type of mechanism or the change in the nature of dominant surface species[54]. Thus, the variation in the Tafel slope can signify a few different underlying possibilities, and the theoretical modeling can help to further elucidate the underlying mechanistic picture.

The predicted Tafel plot on NiOOH (Fig. 4a) shows three distinct linear regions with Tafel slopes of 33, 22, and 50 mV/dec. The first shift of Tafel slope from 33 to 22 mV/dec arises exactly from the transition in the nature of dominant mechanism from LOMa to AEM. The Tafel slope of 22 mV/dec corresponds to the scenario where the concentration of **S1** just starts to rise but is not dominant yet (Figure S4), which is consistent with the result by the classic theoretical analysis of AEM[41]. While the second shift of Tafel slope from 22 to 50 mV/dec arises from the change of dominant surface species to **S1** (Figure S4), and the Tafel slope of 50 mV/dec can be qualitatively understood by the generalized Butler-Volmer (GBV) formalism of Tafel slope[53,55], which indicates that there is one electron-transfer step (starting from the dominant **S1**) before the rate limiting O–O coupling. The Tafel slope of 50 mV/dec in the common range of $j$ matches well with the experimentally reported value[56].

Similarly, on Ni(Co)OOH (Fig. 4b) and Ni(Fe)OOH (Fig. 4c), the shifts of Tafel slopes from 70 to 25 mV/dec and from 66 to 47 mV/dec both arise exactly from the transition in the nature of dominant mechanism from LOMa to AEM. On Ni(Co)OOH, the Tafel slopes of 70 and 25 mV/dec correspond to the scenarios where S0 is the dominant

**Table 1 | Predicted overpotential (η), Tafel slope, and dominant mechanism at η, in comparison with the experimental values**

| Catalyst | Predicted η (mV) | Experiment Tafel slope (mV/dec) | Dominant mechanism at η | η (mV) | Tafel slope (mV/dec) |
|---|---|---|---|---|---|
| NiOOH | 692 | 50 | AEM | 700[24] | 58[56], 93[86] |
| Ni(Co)OOH | 368 | 25 | AEM | 335[87] | 41[87] |
| Ni(Fe)OOH | 293 | 47 | AEM | 300[87] | 35[56], 40[87], 47[88], 48[89] |

surface species (Figure S4) and thus there are one or two electron-transfer steps before the rate limiting O–O coupling step of LOMa and AEM, respectively. It is worth noting that the Co-doping in NiOOH promotes the IMOC-type mechanism that makes a discernible contribution to $j$ (Figure S3c, d) and Tafel slope (Fig. 4b), which is consistent with the findings for OER on CO–Oxide-based catalysts[32]. On Ni(Fe)OOH, the Tafel slopes of 66 and 47 mV/dec correspond to the scenarios where **S0** and **S1** are the dominant surface species (Figure S4), respectively, and thus there is one electron-transfer step before the rate limiting O–O coupling in both regions. The difference between the two Tafel slopes may arise from the different dependences of $\Delta\Omega$ and $\Delta\Omega^{\neq}$ on $\eta_0$ (Table S3).

Table 1 summarizes the predicted $\eta$, Tafel slope, and the dominant mechanism at $\eta$ for OER on each Ni-based LDH from our computational methodology, and the predicted $\eta$ are in excellent agreement with the experimental values as well as the fact that the OER activity follows the order of NiOOH ≪ Ni(Co)OOH < Ni(Fe)OOH. Nevertheless, there is a range of different values reported by experiments for the Tafel slopes, and our predicted Tafel slopes are consistent with some of them, as shown in Table 1. The varying values of Tafel slopes reported by experiments may arise from complications such as the specific adsorptions of particular ions[56,57], and this may compromise the comparison for validating our methodology.

## Discussion

Our results indicate that the LOM-type mechanisms occur on all Ni-based LDHs at observable scales, which is consistent with the isotope-labeling experiment[35]. Although the LOM type of mechanisms can contribute to the OER activity, it consumes lattice oxygens, which may lead to subsequent dissolution of catalyst and thus the instability issue. Thus, our results may explain the superior stability introduced by Fe doping to Ni-based LDHs[10,35] through the suppression of LOM-type mechanisms. The doping of Fe into LDHs promotes AEM for boosted activity, and the promotion of AEM suppresses LOM for improved stability as well. Thus, it may serve as an optimal strategy for improving both activity and stability to promote the AEM type of mechanisms. Also, we find that the Fe-doping quenches the radical character of lattice oxygen as shown in Figure S1, possibly arising from the increased ionic nature of metal-oxygen bonds in the LDH, and this may serve as another strategy to suppress LOM for improved stability.

Nevertheless, the specific means to implement the above strategies may depend strongly on the types of catalysts. Previous studies[45,46] concluded that on the brookite TiO2 and spinel ferrites, the Co sites favor AEM over LOM, so the Co-doping, instead of Fe-doping, may serve as the specific means to promote AEM for improved stability of these OER catalysts. On the perovskites, the contribution of LOM was suggested to be dependent on the covalency of metal-oxygen bonds[58], which may serve as a specific aspect for optimizing the stability.

Moreover, we note that the catalytic performance of Ni(M)OOH for OER is correlated with the spin densities on the M site and O species. Figure S1 shows that the Fe site on Ni(Fe)OOH possesses a large spin population of 2.6 $\mu_B$, and this stabilizes the active *O· via exchange interactions[38], leading to a small $\Delta\Omega$ from **S1** to **S2** with *O· to deliver the O–O coupling in AEM, while the Co and Ni sites on Ni(Co)OOH and NiOOH possess low spin populations of 1.1 and 0.9 $\mu_B$, respectively, resulting in larger $\Delta\Omega$ to overcome for generating the active *O•. Note

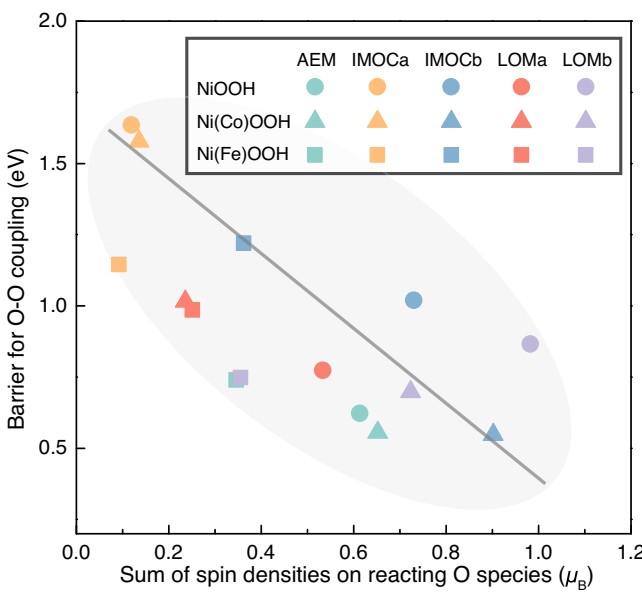

**Fig. 5 | A candidate descriptor for O–O coupling kinetics.** Correlation between the barriers for O–O coupling in all the mechanisms and the sum of spin densities on reacting O species.

that the different magnetic moments can indicate different oxidation states for the TMs (Figure S11). More interestingly, we find that the barriers for O–O coupling in all the mechanisms are well correlated with the spin populations on reacting O species as shown in Fig. 5, which embodies the strength of their radical characters and thus the reactivity. Thus, for optimizing the kinetics of O–O coupling step, higher spin densities on reactive O species are favored, but they generally require higher electrochemical deprotonation energies to produce, so these two considerations may result in a volcano curve for the OER activity with respect to the spin densities on reactive O species. These findings suggest that the OER is a spin sensitive reaction with all types of mechanisms[50], and the spin densities on both the metal site and the active surface O species can serve as delicate descriptors for the OER activity of Ni(M)OOH.

In summary, with combining grand-canonical methods and microkinetic modeling, we elucidate the interplay among the applied potential and competing kinetics in the coupled reaction network composed of all types of mechanisms including AEM, IMOC, and LOM for OER on Ni(M)OOH LDHs, and find that this interplay plays the key role in shaping the contributing mechanisms for OER on LDHs, and results in the potential-dependent transitions of reaction mechanism among distinctive types for OER on all Ni-based LDHs. We also identify that the doping of Fe greatly stabilizes *O· that delivers a more facile AEM and suppresses the LOM-type mechanisms, thus improving the stability of catalyst, and the spin densities on both the metal site and the reactive surface O species can serve as delicate descriptors of the OER activity on Ni(M)OOH.

The predicted overpotentials and Tafel slopes by our methodology match well with the experimental values, and our results also explain the observations in the isotope-labeling experiment, which

arise from the universal presence of LOM-type mechanisms in OER on Ni(M)OOH, although at varied scales with different dopants. Thus, we establish a computational methodology to make accurate predictions and elucidate the potential-dependent mechanisms for OER, as well as other electrochemical reactions, and we expect our methodology to find a wide range of applications in electrochemistry.

## Methods

### Grand-canonical density-functional theory calculations

Density-functional theory (DFT) calculations were carried out with the periodic plane wave framework implemented in the Vienna Ab initio Simulation Package (VASP)[59–61]. The Perdew-Burke-Ernzerhof flavor of the generalized gradient approximation was employed[62], and the Hubbard U correction was included to partly account for the strong correlation in the 3$d$ orbitals of Fe, Co, and Ni[62–64], with the U values of 2.60, 3.50, and 5.20 eV[15], respectively, which have been extensively benchmarked on thermodynamics[65]. The kinetic energy cutoff for the plane-wave basis sets was set to 500 eV (benchmarked as in Figure S14), and the core electrons were represented by the projector augmented wave (PAW) method[66]. The transition states (TSs) were located using the climbing image nudged elastic band (CI-NEB) method[67] to generate the initial guesses, followed by the dimer method[68] to converge to the saddle points.

The solvation effect was included by the implicit electrolyte model implemented in VASPsol[69,70], with a Debye length of 3.0 Å that corresponds to a 1 M ionic strength and a dielectric constant of 78.4 for the aqueous solution, and the applied potential was explicitly modeled with the grand-canonical DFT (GC-DFT) framework[71,72]. This combination of implicit electrolyte model and GC-DFT enables accurate modeling of the electrochemical interface[73–75]. Note that the GC-DFT calculations were performed upon the structures optimized in the implicit electrolyte model to obtain the explicit dependences of free energies on the applied potential. The computational hydrogen electrode (CHE) method was used to describe the free energies of proton–electron pair, with a pH of 14 as the alkaline condition commonly used in the OER on Ni-based LDHs. The calculations of grand free energy profiles and their dependences on $U$ are described in details in SI, and Table S3 lists all the grand free energies at $\eta_0 = 0$ V and their coefficients (dependences) of $U$. More details can be found in SI.

The structural models for the OER-active $\gamma$-phase Ni(M)OOH were constructed with a M/Ni ratio of 1/3 and an average oxidation state of +3.75 to match those in the best-performing catalysts reported by experiments[11,15,29,76–80]. All structures are ferromagnetic and the atomic spin densities were calculated with the Bader scheme[81]. More details and discussions on the structural models can be found in SI and Supplementary Data 1.

### Microkinetic modeling with explicit dependences on the applied potentials

For the microkinetic modeling of reaction network with coupled pathways, we start from the time evolution of coverage for each state $i$,

$$\frac{\partial \theta_i}{\partial t} = \sum_j \left( k_{ji}\theta_j - k_{ij}\theta_i \right) \tag{1}$$

where $k_{ij}$ is the rate constant of elementary reaction from $i$ to $j$, and the summation goes over all states that are connected to $i$ by an elementary step in the reaction network. The steady state regime is reached when all $\theta_i$ do not vary, i.e., $\partial\theta_i/\partial t = 0$, and we can write the master equation in a matrix form to solve[82–84],

$$\hat{k}\boldsymbol{\theta} = 0 \tag{2}$$

Note that the $\hat{k}$ matrix contains a row for the normalization of the $\boldsymbol{\theta}$ vector. Therefore, we can investigate the interplay among all

competing reaction mechanisms via their common states in the above equation. However, when very large current densities are considered, the kinetic Monte Carlo method may be useful to include the coupling to the diffusion process as well as lateral interactions. Nevertheless, we focus on here the region of current density where such complications are insignificant, and the microkinetic modeling method we employ above is accurate for solving the kinetics of coupled reaction network in the steady state regime[82,84].

In addition, the dependence on the applied potential is explicitly introduced in $k_{ij}$, based on the grand-canonical ensemble transition state theory (GCE-TST)[74],

$$k_{ij} = \frac{k_B T}{h} \bullet \exp\left(-\frac{\Delta\Omega^{\neq}}{k_B T}\right) \tag{3}$$

where $k_B$ and $h$ are the Boltzmann and Planck constants, respectively, and $\Delta\Omega^{\neq}$ is the grand free energy barrier that contains the explicit dependence of applied potentials from GC-DFT calculations (see Supplementary Notes in SI). Thus, by solving the master equation $\hat{k}\boldsymbol{\theta} = 0$, the obtained $\boldsymbol{\theta}$ renders the potential-dependent surface coverages of all states (Figure S4).

Note that we employ an approximation for the barriers of electrochemical steps featuring proton transfer, which we justify as follows. We test imposing a rigid minimal barrier ($\Delta\Omega^{\neq}_{min}$) for all the electrochemical steps as described in Figure S9a, and find that $\Delta\Omega^{\neq}_{min}$ has little influence on the kinetics for $\Delta\Omega^{\neq}_{min}$ increasing from 0 to 0.40 eV (Figure S9b). It is worth noting that $\Delta\Omega^{\neq}_{min}$ is an additional part of the barrier, so the barriers of electrochemical steps can be very large in our test, as shown in Figure S9a. We further calculate the barrier for the most endothermic electrochemical step, i.e., from **S1** to **S2** on NiOOH (*OH → *O·), and its $\Delta\Omega^{\neq}_{min}$ is 0.27 eV (Figure S10), which is comparable to the reported value of 0.33 eV for the barriers of electrochemical steps in OER on carbon nanotubes[85]. Thus, we approximate the barriers for the electrochemical steps with their reaction free energies ($\Omega$) when $\Omega > 0$ or zero otherwise, which is equivalent to setting a $\Delta\Omega^{\neq}_{min}$ of 0 eV, and this approximation is also consistent with previous theoretical studies[38,39].

Thus, our microkinetic modeling enables us to calculate the turnover frequency (TOF, $r$) of electrochemical reaction that takes into account the interplay among all competing kinetics in its complex reaction network and the applied potentials. The current density is then calculated as

$$j = \frac{n_e C_e r}{A} \tag{4}$$

where $n_e$ is the number of electrons transferred, $C_e$ is the elementary charge, and $A$ is the surface area per active site. Finally, we can predict the overpotential ($\eta$) as the extra potential required to deliver a $j$ of 10 mA·cm$^{-2}$ with respect to the equilibrium potential. It is worth noting that this microkinetic modeling scheme is less sensitive to the uncertainties in DFT-calculated energetics than the PLS scheme, as shown in Table S4, which may arise from the coupled reaction network we included with kinetic contributions from all types of mechanisms that are complementary to each other.

## Data availability

The data generated in this study are provided in the article and the Supplementary Information files.

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

## Acknowledgements
This work was supported by National Natural Science Foundation of China (Nos. 22122304 and 92261111), Tsinghua University Dushi Program, National Key Research and Development Project (2022YFA1503000), and Tsinghua University Initiative Scientific Research Program (20221080065) all awarded to H.X. We are grateful to the Center of High-Performance Computing at Tsinghua University and Tsinghua Xuetang Talents Program for providing computational resources.

## Author contributions
H.X. conceived and designed the project. Z.W. conducted all the calculations. Z.W., W.A.G., and H.X. analyzed the results. Z.W. wrote the initial manuscript. Z.W., W.A.G., and H.X. revised the manuscript and approved the final version.

## Competing interests
The authors declare no competing interests.
