## [Peer Review File · Nature Communications]

Potential-dependent transition of reaction mechanisms for oxygen evolution on layered double hydroxidesREVIEWER COMMENTS

Reviewer #1 (Remarks to the Author):

the authors present an interesting manuscript on DFT+U simulations of the oxygen evolution reaction (OER) on Ni hydroxide materials including Co and Fe "doped" catalysts. The authors performed calculations of various OER intermediates on surface models and propose a reaction network for the OER mechanism which they use for microkinetic modelling of the reaction. These simulations include a variety of approximations like empirical corrections to DFT, implicit solvation, simple surface models, approximated barriers (not all of them have been computed) and approximated schemes for including the influence of the electrochemical potential. The authors compare their results to experimental findings and discuss the implications of their findings with respect to our understanding of limiting factors in OER catalyst activity and stability.

While this is an interesting and thorough work, the interest for a broader audience beyond electrocatalysis is not really given. While the authors bring up some interesting aspects for scientists who work on water splitting or CO₂ activation in alkaline media, I don't see how this would cater to the audience of Nature Communications. Furthermore, the authors merely apply a combination of existing methods to a very specific type of materials and a very specific process, so that I also don't see a methodological breakthrough that would justify publication in Nature Communications (or really any of the Nature family journals). Hence, I can't recommend publication in Nature Communications and would suggest sending this interesting work to a journal for an appropriate community - physical chemistry, chemical physics or electrocatalysis journals would no doubt be interested in the mechanistic details provided by the authors' work.

Please find further suggestions for a publication in a more focused journal below :

1) The authors should have a thorough look at the literature and cite and discuss alternative approaches and several very similar studies on related problems and materials :

1a) micorkinetic modelling on transition metal oxides for the OER has, for example been done by Dickens et al. with a different scheme to quantify barriers [doi.org/10.1021/acs.jpcc.9b03830] and here, the authors acutaly discuss coverage effects, mention surface Pourbaix diagrams and find that also O₂ desorption can play an important role in the free energy profile (see below). Further work on Fe doped NiOOH was published by Rajan et al. [doi.org/10.1039/D0EE02292F] with a similar approach also including microkinetic modelling and pH as well as potential dependence.

1b) Explicit potential treatments are actually an established methodology which has been used in several other publications by the authors for example for the OER on Ni catalysts in alkaline [doi.org/10.1073/pnas.172203411] or on Ir catalysist in acidic conditiosn [doi.org/10.1021/jacs.6b07557]. Here, a discussion on previous similar work would be helpful.

1c) The OER in alkaline conditions has been discussed by several other groups. A more complete model including surface proubaix diagrams, no a-priori distinction of electrochemical and chemical steps and an explicit treatment of the electrochemical potential for Co and Co/Fe systems was, for example recently published by Bhattacharyya et al. [doi.org/10.1021/acs.jpcc.2c06436] who also investigate spin density, oxidation states and transition state structure for the OER in alkaline conditions.

1c) As far as the surface structures are concerned, by Li et al. (which the authors cite) demonstrates how to incorporate this aspect in simulations on NiOOH [doi.org/10.1021/acscatal.9b04975].

Furthermore, alternative approaches that have been proposed by Gonzalez [doi.org/10.1016/j.jcat.2021.02.026] or Solans-Monfort et al. [doi.org/10.1039/D1NR03592D] nicely demonstrate how DFT simulations can be used to identify the correct surface termination under, for example under acidic conditions and high potentials (see below).

2) I would strongly recommend to compute all barriers rather than estimating them using the BPE approximation. One example would be the LOM on NiOOH. Here, the final step is endothermic and any considerable barrier would have a large influence on the reaction kinetics. However, no barrier that has been calculated using DFT is given, as for most of the other stps.

3) The authors have used the CHE but apply implicit solvation for a shift of the Fermi level to account for the explicit effects of the electrochemical potential on the electronic structure of the material. As the authors stress the importance of the potential in the changes in the reaction network, they should make it clearer (for example by including more figures and tables with the corresponding data) how the energetics change with the potential.

4) The authors use DFT+U and implicit solvation - the reader would be very interested in what the accuracy for relative energies is - typically this is in the order of 0.1 - 0.2 eV, which would result in larger changes in the kinetics.

5) A central aspect in heterogeneous catalysis is Sabatier's principle - if a species or an intermediate binds to the surface too weakly, it will not be adsorbed. If it binds too strongly, it will block the surface and poison the catalyst. In their manuscript, it is not clear to me how the authors have considered surface coverage. While they derive current densities and compute these with some measure of active sites per surface, they have not considered models with different coverages or any parameters that would take this into account.

Furthermore, in electrocatalysis and especially at high oxidizing potentials, the surface state is determined by an interplay of pH and potential. In modern electronic structure simulations of electrocatalytic processes, the first step is the determination of the surface state with respect to coverage with

O, OH and H₂O species. However, the authors do not mention any details how their models correspond to a state of the surface that might be present at experimental conditions.

6) For the Ni/Fe and Ni/Co systems, the authors talk about "doping" - probably because in the experiment, very small amounts for Fe or Co has a large effect on the activity of the Ni catalysts. However, the models the authors use to compute relative energies use relatively small unit cells, resulting in fairly high "doping" concentrations. Here, a more thorough discussion on how realistic and relevant these models are should be included.

7) In the authors list an author has been included that seems not to have contributed to the project design, to the actual computations, the analysis and to writing the manuscript. Nature's publishing policy states (like many other publisher's statements)

"Each author is expected to have made substantial contributions to the conception or design of the work; or the acquisition, analysis, or interpretation of data; or the creation of new software used in the work; or have drafted the work or substantively revised it"

Hence, I would strongly recommend to put any authors that have not contributed significantly to the actual work and results into the acknowledgements.

8) The language and grammar should be checked thoroughly as there are many strange formulations and typos, here are some examples:

I 25 "... half-reactions of great values to energy and ..."

I 92 "Although the OER is a key component to applications of electrochemistry in energy and environment, its sluggish..."

I 84 "...the skeletal formulas of active sites in Figure 1 illustrates on the key..."

I 179 "... it is the AEM that claims the dominated mechanism for OER"

I 235 "... for O-O coupling at low η_0 's "

I 487 references 26 and 30 are identical

Reviewer #2 (Remarks to the Author):

The article entitled "Potential-dependent Transition of Reaction Mechanisms for Oxygen Evolution on Layered Double Hydroxides" by Wang et al. reported a theoretical study on the mechanistic chemistry of OER. The OER is an important multi-electron transfer process, which plays a crucial role in many energy-related applications including electrocatalytic water splitting and metal-air batteries. While many theoretical studies have been conducted to understand atomic insights, most previous studies largely ignored the impact of the working conditions, such as applied bias potential. Indeed, the applied bias potential can significantly affect the properties of the electrode-electrolyte interface. In this study, the authors combined the grand-canonical method with micro-kinetics simulation to fill this theory-experiment gap, which is timely and important to the development of computational electrochemistry. The authors used the LDHs as the model systems to calculate the theoretical overpotential, Tafel sloped. Three different mechanisms were considered. Their results reveal that the applied bias potential can alter the mechanisms. Their computational data can successfully match the experimental observations. It further validates the methodology developed in this study. To this end, I recommend it for publication after addressing the following issues:

- 1) When the authors built the atomic models, they tried to render the oxidation states of metals of +3.75. Since the Ni/Co/Fe cations with such a high oxidation state are often unstable. This oxidation state needs to be validated through either the charge analysis or magnetic moment analysis.
- 2) The magnetic structures may affect the computational results. The authors need to mention whether ferromagnetic or antiferromagnetic structures were used here. What's the change in the magnetic moments and magnetic structures after the adsorption of intermediates under different bias potentials?
- 3) How did the authors calculate the spin densities?
- 4) The authors of VASPsol mentioned that the cut-off energy should be higher than 600 eV. Here, the cut-off energy of 500 eV was used. This parameter needs to be justified.
- 5) In Figure 5, the authors show the spin densities of metals. Some Ni has a spin density of about 1 μ B. However, the spin densities of some of them are almost zero, even in some cases with high symmetry, e.g., HO-Ni-O-Ni-OH. Could the authors explain this difference?

Reviewer #3 (Remarks to the Author):

In the manuscript "Potential-dependent Transition of Reaction Mechanisms for Oxygen Evolution on Layered Double Hydroxides", Wang et al. present a theoretical study of reaction mechanisms for the oxygen evolution reaction (OER) on Ni-based hydroxide electrocatalysts, namely NiOOH, Ni(Co)OOH, and Ni(Fe)OOH. The authors investigate three possible pathways, the adsorbate evolution mechanism (AEM), intramolecular oxygen coupling (IMOC), and lattice oxygen mechanism (LOM). Free energies of intermediates, as well as the transition state of the O-O coupling step, are computed at the level of density-functional theory, and then plugged into a microkinetic model that combines the three reaction pathways. The coupled reaction network is then solved for the steady-state of the various intermediates' surface coverages. This allows the authors to compute the individual contributions of the different reaction mechanisms to the total kinetic current and conclude about the dominant reaction mechanism at different electrode potentials. It is found that the dominant mechanism changes from LOM at small overpotentials to AEM at larger overpotentials. The transition potential from LOM to AEM, as well as the onset potential defined for a current density of 10 mA/cm², depend on the hydroxide composition. The authors claim good agreement with experimental results in terms of the activity trend, onset potentials, and Tafel slopes.

The question of potential-dependent changes in OER mechanisms is highly topical. The authors focus on three commonly investigated mechanisms and the conclusion regarding the dominance of the AEM in the OER-relevant potential range is important. Certain alternatives, however, are neglected, e.g., the coupling of LOM and instability reactions such as cation dissolution. I therefore see the key novelty of the present work in the development of the microkinetic framework for

coupled OER pathways with common intermediates, which I expect to be very useful in future works. The explicit inclusion of DFT-computed free-energy barriers in the microkinetic model is particularly noteworthy. The authors show that this leads to qualitatively different conclusions regarding the dominant mechanism in comparison to the commonly performed analysis of the potential limiting step. The manuscript is well written, albeit repetitive in certain statements. However, relevant parts of the modeling framework should be better justified and presented in more detail to enable the reader adopting the methodology in own future works. In detail, the following points should be addressed before reconsidering the work for publication in Nature Communications.

Methods:

- DFT methods: The authors built their structural models by removing a fraction of interlaminar hydrogen atoms from the beta phase. How realistic are these structures in comparison to experiment? Also, the authors use a Hubbard U correction for 3d electrons. How are the chosen U values justified? I think there should be some comparison of bulk electronic properties, e.g., DOS, with reported experimental results to support the validity of the DFT models.
- CHE method: The authors use the SHE scale in alkaline conditions. Although this is not wrong, it is slightly confusing and makes it difficult to assess the method employed. I suggest to convert to the RHE scale by combining the SHE scale with the pH correction. The presentation of the CHE method in section 2 of the SI should be made clearer. As given, the equations in this section appear to be inconsistent, e.g., there is no potential dependence in the equation for ΔG at line 55. In the CHE equation at line 45, the authors state that the applied potential is referenced to the SHE. I think it must be rather the applied potential vs. RHE (alkaline), because, at $U = 0$ V vs. RHE, the hydrogen electrode reaction must be at equilibrium at any pH, so $G(\text{H}^+) + G(\square\text{e}^-) = 1/2 G(\text{H}_2)$. Likewise, I'm confused by the value of 0.401 eV per transferred electron used in the equation at line 91 of the SI, which apparently corresponds to the SHE reference. Converting to RHE and using a value of 1.23 eV per transferred electron would be easier to understand.
- How were the free energies of molecular species (H_2O , H_2 , O_2) obtained, as given in Table S2?
- Grand-canonical DFT method: What is the value of the Debye length and other relevant VASPsol parameters? The authors assume a linearized relationship between the grand potential Ω and the electrode potential U . However, this neglects capacitive effects, since the interfacial capacitance defines the curvature of the $\Omega(U)$ curve. Is it justified to neglect the capacitive effect? Finally, how was the GC-DFT method combined with NEB to obtain the transition state under constant potential conditions?
- Microkinetic modeling: I am concerned about the pre-exponential factor used in the expression for the rate constants at line 380. The authors simply use the typical (kT/h) factor. However, additional pre-factors should be needed to account for the correct normalization of the rate constants with respect to the reference state used for $\Delta\Omega$. Particularly, the rate equations at line 365 explicitly include only the coverages of surface intermediates, but the reaction steps also involve non-adsorbed reactants, namely H_2O , OH^- etc. Apparently, the respective concentrations of non-adsorbed species are lumped into the rate constants, which, in principle, is ok if these concentrations are constant. However, this must affect the respective pre-exponential factors, so I'm surprised about the simple (kT/h) used by the authors. In my opinion, the expression for the rate constants must be fundamentally motivated and explicitly related to the form of the rate equations at line 365 and the CHE reference for the DFT-computed $\Delta\Omega$.
- Electrochemical steps: Figure S9: The authors appear to assume that the barrier height changes 1:1 with the free-energy change between initial and final state. Effectively, this corresponds to a symmetry factor of $\beta = 1$. Typically, symmetry factors of electrochemical steps are rather around 1/2. How is the assumption of $\beta = 1$ justified?

Results:

- AEM: The O-O coupling step is considered with H₂O as reactant. Alternatively, OH⁻ could serve as a reactant for this step in alkaline conditions. How would the results be affected by this choice? Is the microkinetic model able to discern these two alternatives and predict the dominant one?

- Comparison with experiment (Table 1): For each material, the authors use the experimental overpotential from only one respective study for comparison with their modeling results. There are many experimental reports on NiOOH-based OER catalysts, so a more extensive literature comparison would be welcome. Regarding Tafel slopes, given the scattering of experimentally reported values, are they really useful for validating the model results?

Presentation:

- Several figures (in manuscript and SI) are difficult to read due to small size and poor resolution. The resolution and readability should be improved.

- Lines 165-167: This sentence is difficult to understand, please rephrase. What do the authors mean with the "PLS is not turned on"?

Point-by-point Response to the Reviewers' Comments

General Remarks: We are deeply grateful to all the reviewers for their critical and constructive comments, which contribute to improving the quality of this manuscript. We have carefully addressed all the comments and revised the manuscript accordingly. **The revisions are highlighted in the manuscript.** The point-by-point responses are listed below, with the reviewers' original comments shown in italics and our responses shown in blue.

Reviewer #1

the authors present an interesting manuscript on DFT+U simulations of the oxygen evolution reaction (OER) on Ni hydroxide materials including Co and Fe "doped" catalysts. The authors performed calculations of various OER intermediates on surface models and propose a reaction network for the OER mechanism which they use for microkinetic modelling of the reaction. These simulations include a variety of approximations like empirical corrections to DFT, implicit solvation, simple surface models, approximated barriers (not all of them have been computed) and approximated schemes for including the influence of the electrochemical potential. The authors compare their results to experimental findings and discuss the implications of their findings with respect to our understanding of limiting factors in OER catalyst activity and stability.

While this is an interesting and thorough work, the interest for a broader audience beyond electrocatalysis is not really given. While the authors bring up some interesting aspects for scientists who work on water splitting or CO₂ activation in alkaline media, I don't see how this would cater to the audience of Nature Communications. Furthermore, the authors merely apply a combination of existing methods to a very specific type of materials and a very specific process, so that I also don't see a methodological breakthrough that would justify publication in Nature Communications (or really any of the Nature family journals). Hence, I can't recommend publication in Nature Communications and would suggest sending this interesting work to a journal for an appropriate community - physical chemistry, chemical physics or electrocatalysis journals would no doubt be interested in the mechanistic details provided by the authors' work.

Response: We are grateful to the reviewer for finding our work interesting and thorough, but we respectfully disagree with the reviewer's assessment of the significance and novelty of our work.

Regarding the significance, OER is a universal anodic half-reaction that is essential for applications of electrochemistry in energy and environment, and a thorough understanding of its mechanism is key to the rational design of high-performing OER electrocatalysts for practical applications.

Moreover, the OER mechanism is of fundamental scientific interest, because it involves the oxidation of water molecule into O₂ that requires a large electrochemical driving force, and it is intriguing to understand how the O-O coupling is delivered and how it depends on the applied potential.

Consequently, there have been numerous studies devoted to understanding the OER mechanisms, to name but a few,

- Key role of chemistry versus bias in electrocatalytic oxygen evolution. *Nature* **587**, 408-413 (2020).
- In-situ structure and catalytic mechanism of NiFe and CoFe layered double hydroxides during oxygen evolution. *Nat. Commun.* **11**, 2522 (2020).
- Direct evidence of boosted oxygen evolution over perovskite by enhanced lattice oxygen participation. *Nat. Commun.* **11**, 2002 (2020).
- Correlative operando microscopy of oxygen evolution electrocatalysts. *Nature* **593**, 67-73 (2021).
- Mechanistic insight into the active centers of single/dual-atom Ni/Fe-based oxygen electrocatalysts. *Nat. Commun.* **12**, 5589 (2021).
- Spin-polarized oxygen evolution reaction under magnetic field. *Nat. Commun.* **12**, 2608 (2021).
- Pivotal role of reversible NiO₆ geometric conversion in oxygen evolution. *Nature* **611**, 702-708 (2022).
- Free energy difference to create the M-OH* intermediate of the oxygen evolution reaction by time-resolved optical spectroscopy. *Nat. Mater.* **21**, 88-94 (2022).
- Spectroelectrochemical Analysis of the Water Oxidation Mechanism on Doped Nickel Oxides. *J. Am. Chem. Soc.* **144**, 7622-7633 (2022).

These studies (and many more) were well-recognized by a broad audience, so we can conclude that this topic, the fundamental understanding of OER mechanisms, is a perfect fit to journals like *Nature Communications*. Our work unravels a novel understanding of OER mechanisms, *i.e.*, the potential-dependent transitions in the nature of mechanisms, and thus we believe that it is of broad interest to the readership of *Nature Communications*.

Regarding the novelty in the methodology, our work recognizes that the different types of OER mechanisms previously proposed are in fact idealizations of a single coupled reaction network with competing kinetics that are strongly potential-dependent, so we combined a set of suitable methods to explicitly investigate this coupled reaction network and its potential-dependent kinetics. This led us to the novel finding that the nature of OER mechanism can change between distinctive types as a function of the applied potential. Therefore, **we consider the recognition and investigation of the coupled reaction network and its potential-dependent kinetics in our work to be a major advance in the methodology, *i.e.*, the way how we model the electrochemical reactions, which can reveal novel aspects of electrochemical reactions.**

The literature suggested by the reviewer in Comment #1 below also strongly demonstrates that our work is a major advance. None of these previous studies were able to recognize the coupled reaction network and uncover the potential-dependent transition in the nature of OER mechanism.

This advance by our work can be applicable to other electrochemical reactions, which often have a reaction network coupled with the applied potential, including the seemingly simple HER, as we discussed in the manuscript. This can spark a wide range of investigations that may lead to many more novel understandings. Therefore, again, we believe that our work is of broad interest to the readership of *Nature Communications*.

To emphasize these points, we have revised the manuscript accordingly (Pages 2, 3, 4).

1. The authors should have a thorough look at the literature and cite and discuss alternative approaches and several very similar studies on related problems and materials :

1a) *micorkinetic modelling on transition metal oxides for the OER has, for example been done by Dickens et al. with a different scheme to quantify barriers [doi.org/10.1021/acs.jpcc.9b03830] and here, the authors acutaly discuss coverage effects, mention surface Pourbaix diagrams and find that also O2 desorption can play an important role in the free energy profile (see below). Further work on Fe doped NiOOH was published by Rajan et al. [doi.org/10.1039/D0EE02292F] with a similar approach also including microkinetic modelling and pH as well as potential dependence.*

1b) *Explicit potential treatments are actually an established methodology which has been used in several other publications by the authors for example for the OER on Ni catalysts in alkaline [doi.org/10.1073/pnas.172203411] or on Ir catalysist in acidic conditiosn [doi.org/10.1021/jacs.6b07557]. Here, a discussion on previous similar work would be helpful.*

1c) *The OER in alkaline conditions has been discussed by several other groups. A more complete model including surface proubaix diagrams, no a-priori distinction of electrochemical and chemical steps and an explicit treatment of the electrochemical potential for Co and Co/Fe systems was, for example recently published by Bhattacharyya et al. [doi.org/10.1021/acs.jpcc.2c06436] who also investigate spin density, oxidation states and transition state structure for the OER in alkaline conditions.*

1d) *As far as the surface structures are concerned, by Li et al. (which the authors cite) demonstrates how to incorporate this aspect in simulations on NiOOH [doi.org/10.1021/acscatal.9b04975]. Furthermore, alternative approaches that have been proposed by Gonzalez [doi.org/10.1016/j.jcat.2021.02.026] or Solans-Monfort et al. [doi.org/10.1039/D1NR03592D] nicely demonstrate how DFT simulations can be used to identify the correct surface termination under, for example under acidic conditions and high potentials (see below).*

Response: We thank the reviewer for pointing us to a set of relevant literature, and we have included them in the corresponding discussions (Page 3). However, as we discussed in the previous response, we must point out that none of these previous studies (including ours) were able to recognize the coupled reaction network composed of different types of OER mechanisms, and thus none of them discovered the potential-dependent transition in the nature of OER mechanism.

2. I would strongly recommend to compute all barriers rather than estimating them using the BPE approximation. One example would be the LOM on NiOOH. Here, the final step is endothermic and any considerable barrier would have a large influence on the reaction kinetics. However, no barrier that has been calculated using DFT is given, as for most of the other steps.

Response: We thank the reviewer for this constructive suggestion, which helps to highlight one key point in our methodology.

First, we did not use the BEP relation to estimate the barriers. Instead, we explicitly calculated all the barriers for the O-O coupling chemical steps in all types of mechanisms by locating the transition states. For the electrochemical steps featuring proton transfer, we justified an approximation of their barriers for calculating the kinetics, as described in the Methods section and Figures S9-S10 in the SI.

Our justification of the approximation is as follows.

We test imposing a rigid minimal barrier ($\Delta\Omega_{\min}^{\ddagger}$) for all the electrochemical steps as described in Figure S9a (Figure S9 is shown below as **Figure R1** for the reviewer's convenience) and find that $\Delta\Omega_{\min}^{\ddagger}$ has little influence on the kinetics for $\Delta\Omega_{\min}^{\ddagger}$ increasing from 0 to 0.40 eV, as shown in Figure S9b. It is worth noting that $\Delta\Omega_{\min}^{\ddagger}$ is an additional part of the barrier, so the barriers of electrochemical steps can be very large in our test, as shown in Figure S9a.

To further justify that $\Delta\Omega_{\min}^{\ddagger}$ is smaller than 0.40 eV for electrochemical steps in OER, we explicitly calculated the barrier for the most endothermic electrochemical step (from S1 to S2 on NiOOH), which is much more endothermic than the final step in LOM on NiOOH as suggested by the reviewer, and we found that $\Delta\Omega_{\min}^{\ddagger}$ for this step is only 0.27 eV (Figure S10).

Thus, based on our test shown in Figure S9 (Figure R1), we approximated all the barriers of electrochemical steps featuring proton transfer with $\Delta\Omega_{\min}^{\ddagger} = 0$ eV, *i.e.*, their barriers are their reaction free energies ($\Delta\Omega$) when $\Delta\Omega > 0$ or zero otherwise.

Therefore, with the approximation above and the explicitly calculated barriers for the O-O coupling chemical steps, we included the kinetic contributions from all steps to construct the \hat{k} matrix for microkinetic modelling.

Figure R1 (Figure S9). (a) Illustration of the rigid minimal barrier ($\Delta\Omega_{\min}^{\ddagger}$) of 0.40 eV we impose on the barrier of electrochemical step with $\Delta\Omega = 0$ eV at $U = 0.8$ V. The barrier ($\Delta\Omega^{\ddagger}$) of electrochemical step is related to $\Delta\Omega_{\min}^{\ddagger}$ and the reaction free energy ($\Delta\Omega$) via the expression below,

$$\Delta\Omega^{\ddagger} = \begin{cases} \Delta\Omega_{\min}^{\ddagger}, & \text{for } \Delta\Omega \leq 0 \\ \Delta\Omega_{\min}^{\ddagger} + \beta \cdot \Delta\Omega & \text{for } \Delta\Omega > 0 \end{cases}$$

So, $\Delta\Omega_{\min}^{\ddagger}$ sets the lower bound for the barrier when $\Delta\Omega \leq 0$ and becomes an additional rigid term

to the barrier when $\Delta\Omega > 0$. In order to test the most extreme scenario, we chose $\beta = 1$ here, because this gives the fastest increase in $\Delta\Omega^\ddagger$ as $\Delta\Omega$ increases (when $\Delta\Omega > 0$). (Note that β is typically ~ 0.5 , so we also performed a test with $\beta = 0.5$ in Figure S16 that gives the same conclusions.) (b) Current density as the function of overpotential η_0 ($U - 401$ mV) with different rigid minimal barrier ($\Delta\Omega_{\min}^\ddagger$) imposed on the electrochemical steps in OER on NiOOH. Note that the current density and thus the catalytic performance has little dependence on the $\Delta\Omega_{\min}^\ddagger$ of electrochemical step up to 0.4 eV. (Please refer to Figure S9 in the SI for a more detailed caption.)

We believe that our justification and approximation would be of great interest to the community of theoretical and computational electrochemistry. It is also consistent with the previous theoretical study by Dickens *et al.* [*J. Phys. Chem. C* 2019, 123, 18960-18977] suggested by the reviewer in Comment #1, which concluded that “**the kinetics of proton transfer ... is facile and that O-O bond formation is most likely rate-determining in all cases.**” Our previous study [*Proc. Natl. Acad. Sci. U.S.A.* 2018, 115, 5872-5877] also employed this approximation, although not fully justified as in our current work. To emphasize this, we have revised the manuscript (Page 23) accordingly.

3. The authors have used the CHE but apply implicit solvation for a shift of the Fermi level to account for the explicit effects of the electrochemical potential on the electronic structure of the material. As the authors stress the importance of the potential in the changes in the reaction network, they should make it clearer (for example by including more figures and tables with the corresponding data) how the energetics change with the potential.

Response: We thank the reviewer for this constructive suggestion. In the Additional Computational Details section in the SI, we included a part (Part 5 on Pages S6-S7) that describes in details how we calculated the grand free energy profiles and their dependences on the applied potential. In addition, we included Table S3 (Page S10) that lists all the grand free energies at $\eta_0 = 0$ V and their coefficients (dependences) of U , *i.e.*, how they change with U . To make it clearer, we have revised the Method section of the manuscript (Page 21) to point the readers to these details in the SI more specifically.

4. The authors use DFT+U and implicit solvation - the reader would be very interested in what the accuracy for relative energies is - typically this is in the order of 0.1 - 0.2 eV, which would result in larger changes in the kinetics.

Response: We thank the reviewer for this intriguing comment.

First, benchmarking the accuracy of DFT methods for energetics in heterogeneous electrochemistry is challenging, because there are neither affordable high-level *ab initio* methods (such as coupled-cluster or multi-reference methods) for modeling heterogeneous electrochemistry to provide the accurate references nor abundant fundamental energetics (such as adsorption energies under electrochemical conditions) from experiments. Currently, we believe that benchmarking against experimental overpotentials and Tafel slopes is a viable way to justify the theoretical method, and our results summarized in Table 1 show that our methodology is sufficiently accurate for predicting these experimental quantities, which justifies the suitability of our methodology for heterogeneous electrochemistry.

Second, inspired by this comment, we tested how much the possible deviation of calculated energetics affects the predicted overpotentials as follows.

We changed the free energy of every state by ± 0.1 eV to mimic the uncertainty in DFT-calculated energetics, and calculated the overpotentials via our microkinetic modeling scheme as well as via the PLS scheme. **Table R1** below lists the changes in the predicted overpotentials induced by the introduced deviation (ϵ) of ± 0.1 eV in the free energy of each state. The results show that the overpotential predicted by our microkinetic modeling scheme is less sensitive to the uncertainties in DFT-calculated energetics than the PLS scheme. The predicted overpotential changes by at most 0.08 V and is insensitive to ϵ in the free energies of most states. This may arise from the coupled reaction network we included, which has the kinetic contributions from all types of mechanisms that are complementary to each other.

We have added **Table R1** as Table S4 in the SI and revised the manuscript (Page 23) to point the readers to this in the SI.

Table R1. Changes in the predicted overpotential ($\Delta\eta$) induced by the introduced deviation (ε) of ± 0.1 eV in the free energy of each state to mimic the uncertainty in DFT-calculated energetics. (Note that “N. A.” marks the states that are not present in the PLS scheme.)

State	Microkinetic modeling scheme		PLS scheme	
	$\varepsilon = +0.10$ eV	$\varepsilon = -0.10$ eV	$\varepsilon = +0.1$ eV	$\varepsilon = +0.10$ eV
	$\Delta\eta$ (V)			
S0	0	0	S0	0
S1	-0.08	0.08	S1	-0.08
S2	0	0.01	S2	0
S3	0	0	S3	0
S4	0	0	S4	0
S5	0	0	S5	0
S6	0	0	S6	0
S7	0	0	S7	0
S8	0	0	S8	0
S9	0	0	S9	0
TS_AEM	0.07	-0.08	TS_AEM	0.07
TS_IMOCa	0	0	TS_IMOCa	0
TS_IMOCb	0	0	TS_IMOCb	0
TS_LOMa	0	0	TS_LOMa	0
TS_LOMb	0	-0.03	TS_LOMb	0

5. A central aspect in heterogeneous catalysis is Sabatier's principle - if a species or an intermediate binds to the surface too weakly, it will not be adsorbed. If it binds too strongly, it will block the surface and poison the catalyst. In their manuscript, it is not clear to me how the authors have considered surfaces coverage. While they derive current densities and compute these with some measure of active sites per surface, they have not considered models with different coverages or any parameters that would take this into account.

Furthermore, in electrocatalysis and especially at high oxidizing potentials, the surface state is determined by an interplay of pH and potential. In modern electronic structure simulations of electrocatalytic processes, the first step is the determination of the surface state with respect to

coverage with O, OH and H₂O species. However, the authors do not mention any details how their models correspond to a state of the surface that might be present at experimental conditions.

Response: We thank the reviewer for this constructive comment.

It is exactly the microkinetic modeling that was employed to quantitatively confirm the Sabatier's principle, such as in the classic study by Vojvodic *et al.* [*Chem. Phys. Lett.* 2014, 598, 108-112]. And the microkinetic modeling includes the surface coverages of all states by design.

The microkinetic modeling scheme used in our work was described in details in the Methods section. The rate constants in the \hat{k} matrix are explicitly dependent on the applied potential, and thus by solving the master equation $\hat{k}\vec{\theta} = 0$, the obtained $\vec{\theta}$ vector contains the potential-dependent surface coverages of all states (i.e., the surface Pourbaix diagram), which were provided in Figure S4.

To emphasize this, we have revised the manuscript (Page 22) to point the readers to Figure S4 in SI.

6. For the Ni/Fe and Ni/Co systems, the authors talk about "doping" - probably because in the experiment, very small amounts for Fe or Co has a large effect on the activity of the Ni catalysts. However, the models the authors use to compute relative energies use relatively small unit cells, resulting in fairly high "doping" concentrations. Here, a more thorough discussion on how realistic and relevant these models are should be included.

Response: We thank the reviewer for this constructive comment. In our work, we used the experimentally reported M/Ni ratio in the best-performing Ni(M)OOH catalysts, which is 1/3 [from the references we cited: *Adv. Energy Mater.* **6**, 1600621 (2016); *Chem. Soc. Rev.* **46**, 337-365 (2017); *Nat. Commun.* **11**, 2522 (2020)], and this, together with the construction of models, was described in details in Part 1 (Pages S2-S3) of the Additional Computational Details section in the SI. To clarify this, we have added a brief description to the Methods section in the manuscript (Page 21) and point the readers to more details in the SI.

7. In the authors list an author has been included that seems not to have contributed to the project design, to the actual computations, the analysis and to writing the manuscript. Nature's publishing policy states (like many other publisher's statements)

"Each author is expected to have made substantial contributions to the conception or design of the work; or the acquisition, analysis, or interpretation of data; or the creation of new software used in the work; or have drafted the work or substantively revised it"

Hence, I would strongly recommend to put any authors that have not contributed significantly to the actual work and results into the acknowledgements.

Response: We thank the reviewer for pointing out this omission. All authors made substantial contributions to analyzing the results and revising the manuscript. We have revised the Author information section (Page 24) to properly acknowledge the contributions from all authors.

8. The language and grammar should be checked thoroughly as there are many strange formulations and typos, here are

some examples:

l 25 "... half-reactions of great values to energy and ..."

l 92 "Although the OER is a key component to applications of electrochemistry in energy and environment, its sluggish..."

l 84 "..the skeletal formulas of active sites in Figure 1 illustrates on the key..."

l 179 "... it is the AEM that claims the dominated mechanism for OER"

l 235 "... for O-O coupling at low η_0 's "

l 487 references 26 and 30 are identical

Response: We thank the reviewer for this careful examination and constructive suggestions. We have carefully polished the manuscript and tidied up the references.

Reviewer #2

The article entitled "Potential-dependent Transition of Reaction Mechanisms for Oxygen Evolution on Layered Double Hydroxides" by Wang et al. reported a theoretical study on the mechanistic chemistry of OER. The OER is an important multi-electron transfer process, which plays a crucial role in many energy-related applications including electrocatalytic water splitting and metal-air batteries. While many theoretical studies have been conducted to understand atomic insights, most previous studies largely ignored the impact of the working conditions, such as applied bias potential. Indeed, the applied bias potential can significantly affect the properties of the electrode-electrolyte interface. In this study, the authors combined the grand-canonical method with micro-kinetics simulation to fill this theory-experiment gap, which is timely and important to the development of computational electrochemistry. The authors used the LDHs as the model systems to calculate the theoretical overpotential, Tafel sloped. Three different mechanisms were considered. Their results reveal that the applied bias potential can alter the mechanisms. Their computational data can successfully match the experimental observations. It further validates the methodology developed in this study. To this end, I recommend it for publication after addressing the following issues:

Response: We are deeply grateful to the reviewer for the thorough evaluation and highly positive comments on our work, and we have carefully addressed the reviewer's specific comments as follows.

1. When the authors built the atomic models, they tried to render the oxidation states of metals of +3.75. Since the Ni/Co/Fe cations with such a high oxidation state are often unstable. This oxidation state needs to be validated through either the charge analysis or magnetic moment analysis.

Response: We thank the reviewer for this constructive and expert comment.

In this work, we focused on the average oxidation state (OS), which is the quantity reported by experimental characterizations and can be derived from the stoichiometry. In order to validate it, we have performed the magnetic moment analysis of the NiOOH matrix as suggested by the reviewer. The results (**Figure R2** below) show that, at η of 0.7 V (the working potential for OER on NiOOH),

- the stoichiometry is $\text{Ni}_8\text{O}_{16}\text{H}_2$, which renders an average OS of +3.75 for Ni;
- the total magnetic moment is $2 \mu_{\text{B}}$, which are localized on two Ni sites (shown by the spin density

contour in the inset of **Figure R2**), indicating that there are two Ni^{3+} (one unpaired electron) and six Ni^{4+} (no unpaired electron), thus confirming an average OS of +3.75.

Based on this NiOOH matrix with the average OS of +3.75, we built the Fe- and Co-doped cases and their surfaces. To further confirm that the catalyst surfaces can reach the average OS of +3.75 at working conditions, we calculated the grand-canonical free energy changes for the electrochemical deprotonation (oxidation) of all the surface models as shown in **Figure R3** below, and the results indicate that it becomes thermodynamically favorable (exothermic) to deprotonate (oxidize) the surfaces to reach the average OS of +3.75 at the working potentials for OER.

To clarify this, we have added **Figures R2** and **R3** into the SI as Figures S11 and S12, and we have revised the manuscript (Page 21) and SI (Page S3) accordingly.

Figure R2. The average oxidation state (OS) and the total magnetic moment of gradually deprotonated NiOOH with n denoting the number of H atoms and η the corresponding overpotential. The insets are the spin density contours of the systems. The electron configurations of Ni^{2+} , Ni^{3+} , and Ni^{4+} are shown in the right panel, which indicate their numbers of unpaired electron(s).

Figure R3. The grand-canonical free energy changes ($\Delta\Omega$) for the electrochemical deprotonation (oxidation) of the surface models to reach the average OS of +3.75. Note that η_H marks the applied potential when $\Delta\Omega = 0$, and all η_H are lower than the corresponding η (the working potential for OER on each catalyst), so it is exothermic for all surfaces to reach the average OS of +3.75 at η .

2. The magnetic structures may affect the computational results. The authors need to mention whether ferromagnetic or antiferromagnetic structures were used here. What's the change in the magnetic moments and magnetic structures after the adsorption of intermediates under different bias potentials?

Response: We thank the reviewer for this constructive and expert comment.

All the structures are ferromagnetic and remain so after the adsorptions of intermediates under different potentials. In order to provide more details about the magnetic structures at different potentials, we have added Table S5 (shown below as **Table R2**), which lists the total magnetic moments for all states at different potentials, and Figures S13 (shown below as **Figure R4**), which shows the atomic spin densities of catalytically active metal sites in the states before the O-O coupling chemical steps.

To clarify this, we have revised the manuscript (Page 21) and SI (Page S3) accordingly to point the readers to these details in the SI.

Table R2. Total magnetic moments (μ_B) for all states at different potentials.

	NiOOH		Ni(Co)OOH		Ni(Fe)OOH	
	$U=0$	$U=2\text{ V}$	$U=0$	$U=2\text{ V}$	$U=0$	$U=2\text{ V}$
S0	7.33	7.28	10.82	10.69	17.16	16.78
S1	7.58	7.80	8.53	8.33	19.74	19.75
S2	7.69	8.74	9.96	10.57	18.95	18.79
S3	7.37	8.66	8.69	8.82	18.70	18.85
S4	7.60	8.18	9.84	9.13	19.70	19.95
S5	8.97	8.10	7.74	8.11	21.52	22.09
S6	8.21	8.31	8.89	11.00	14.17	14.84
S7	8.07	8.10	8.70	10.71	19.66	19.89
S8	9.18	8.83	8.91	12.99	22.47	22.92
S9	8.12	8.02	9.63	7.94	19.58	19.92

Figure R4. Atomic spin densities of catalytically active metal sites and reacting O species in the states of S1 and S2 for (a, b) NiOOH, (c, d) Ni(Co)OOH, and (e, f) Ni(Fe)OOH under different potentials.

3. How did the authors calculate the spin densities?

Response: We thank the reviewer for this expert question. The atomic spin densities shown in Figure S1 were calculated with the Bader scheme. To clarify this, we have revised the manuscript (Page 21) accordingly.

4. The authors of VASPsol mentioned that the cut-off energy should be higher than 600 eV. Here, the cut-off energy of 500 eV was used. This parameter needs to be justified.

Response: We thank the reviewer for this constructive and expert comment.

As shown in **Figure R5** below, we have performed benchmark VASPsol calculations on the cut-off energy with all the reaction energies in AEM on NiOOH by increasing the cut-off energy from 400 to 600 eV at an interval of 50 eV. The results suggest that the cut-off energy of 500 eV is sufficient to converge the reaction energies to within 0.005 eV, which is less than the typical uncertainty in the DFT-calculated energetics. This justifies the cut-off energy of 500 eV used in this work. To clarify this, we have added **Figure R5** as Figure S14 in the SI and revised the manuscript (Page 20) accordingly.

Figure R5. Benchmark calculations on the cut-off energy with all the reaction energies in AEM on

NiOOH. This shows that the cut-off energy of 500 eV is sufficient to converge the reaction energies to within 0.005 eV, which is less than the typical uncertainty in the DFT-calculated energetics.

5. In Figure 5, the authors show the spin densities of metals. Some Ni has a spin density of about 1 μ_B . However, the spin densities of some of them are almost zero, even in some cases with high symmetry, e.g., HO-Ni-O-Ni-OH. Could the authors explain this difference?

Response: We thank the reviewer for this careful examination and inspiring question.

As discussed in the response to Comment #1, the NiOOH catalyst at the working potential contains mixed Ni³⁺ and Ni⁴⁺ sites, resulting an average OS of +3.75. The Ni³⁺ site has a magnetic moment of 1 μ_B , while the Ni⁴⁺ site has no magnetic moment, as shown in **Figure R2**. Thus, the different magnetic moments on Ni indicate different oxidation states for Ni.

The local high symmetry of HO-Ni-O-Ni-OH in the S1 state is globally broken by the H atoms on the neighboring bridge O site and lattice O site, and thus this leads to the different magnetic moments of 1 and 0 μ_B that indicate a Ni³⁺ site and a Ni⁴⁺ site, respectively.

To clarify this, we have revised the manuscript accordingly (Page 19).

Reviewer #3

In the manuscript "Potential-dependent Transition of Reaction Mechanisms for Oxygen Evolution on Layered Double Hydroxides", Wang et al. present a theoretical study of reaction mechanisms for the oxygen evolution reaction (OER) on Ni-based hydroxide electrocatalysts, namely NiOOH, Ni(Co)OOH, and Ni(Fe)OOH. The authors investigate three possible pathways, the adsorbate evolution mechanism (AEM), intramolecular oxygen coupling (IMOC), and lattice oxygen mechanism (LOM). Free energies of intermediates, as well as the transition state of the O-O coupling step, are computed at the level of density-functional theory, and then plugged into a microkinetic model that combines the three reaction pathways. The coupled reaction network is then solved for the steady-state of the various intermediates' surface coverages. This allows the authors to compute the individual contributions of the different reaction mechanisms to the total kinetic current and conclude about the dominant reaction mechanism at different electrode potentials. It is found that the dominant mechanism changes from LOM at small overpotentials to AEM at larger overpotentials. The transition potential from LOM to AEM, as well as the onset potential defined for a current density of 10 mA/cm², depend on the hydroxide composition. The authors claim good agreement with experimental results in terms of the activity trend, onset potentials, and Tafel slopes.

The question of potential-dependent changes in OER mechanisms is highly topical. The authors focus on three commonly investigated mechanisms and the conclusion regarding the dominance of the AEM in the OER-relevant potential range is important. Certain alternatives, however, are neglected, e.g., the coupling of LOM and instability reactions such as cation dissolution. I therefore see the key novelty of the present work in the development of the microkinetic framework for coupled OER pathways with common intermediates, which I expect to be very useful in future works. The explicit inclusion of DFT-computed free-energy barriers in the microkinetic model is particularly noteworthy. The authors show that this leads to qualitatively different conclusions regarding the dominant mechanism in comparison to the commonly performed analysis of the potential limiting step. The manuscript is well written, albeit repetitive in certain statements. However, relevant parts of the modeling framework should be better justified and presented in more detail to enable the reader adopting the methodology in own future works. In detail, the following points should be addressed before reconsidering the work for publication in Nature Communications.

Response: We are deeply grateful to the reviewer for the thorough evaluation and highly positive comments on our work, and we have carefully addressed the reviewer's specific comments as follows.

1. DFT methods: The authors built their structural models by removing a fraction of interlaminar hydrogen atoms from the beta phase. How realistic are these structures in comparison to experiment? Also, the authors use a Hubbard U correction for 3d electrons. How are the chosen U values justified? I think there should be some comparison of bulk electronic properties, e.g., DOS, with reported experimental results to support the validity of the DFT models.

Response: We thank the reviewer for this constructive and expert comment.

Regarding the structural models, the OER-active γ -phase of Ni(M)OOH is of low crystallinity and high structural complexity, so the experiments cannot provide well-determined structures for it but only information on the average oxidation state (OS) of Ni(M) and the M/Ni ratio. Consequently, we built the structural models by starting from the well-defined β -phase, which is the precursor of γ -phase in experiments, and deprotonating it to reach the average OS of +3.75 to match the experiments.

In addition, the γ -phase contains intercalated species (H_2O and cations) that change the interlayer distance, so we also checked the influence of the interlayer distance in our model, as shown in Figure S5. We changed the interlayer distance from 4.87 Å to 10 Å and found that the reaction free energies do not vary much (within 0.13 eV). Note that the effects of intercalated H_2O and cations are taken into account by the implicit electrolyte model.

Therefore, we believe that our models serve as suitable representations of the working catalysts in experiments, which is also indirectly supported by the predicted overpotentials and Tafel slopes that match well with the experiments.

To clarify this, we have revised the manuscript (Page 21) and SI (Page S2) accordingly.

Regarding the Hubbard U correction, we used the U values from [*Nat. Commun.* **11**, 2522 (2020)], which actually have been extensively benchmarked on the thermodynamics of redox and (de)hydration for a series of bulk transition metal (Fe, Co, and Ni) hydroxides, oxyhydroxides, binary, and ternary oxides [*J. Phys. Chem. C* **119**, 18177–18187 (2015)], and the standard error with respect to the experimental values is only 0.04 eV per reaction formula unit. This justifies our use of these U values

for calculating the free energy profiles.

To clarify this, we have revised the manuscript (Page 20) accordingly.

2. CHE method: The authors use the SHE scale in alkaline conditions. Although this is not wrong, it is slightly confusing and makes it difficult to assess the method employed. I suggest to convert to the RHE scale by combining the SHE scale with the pH correction. The presentation of the CHE method in section 2 of the SI should be made clearer. As given, the equations in this section appear to be inconsistent, e.g., there is no potential dependence in the equation for ΔG at line 55. In the CHE equation at line 45, the authors state that the applied potential is referenced to the SHE. I think it must be rather the applied potential vs. RHE (alkaline), because, at $U = 0$ V vs. RHE, the hydrogen electrode reaction must be at equilibrium at any pH, so $G(H^+) + G(e^-) = 1/2 G(H_2)$. Likewise, I'm confused by the value of 0.401 eV per transferred electron used in the equation at line 91 of the SI, which apparently corresponds to the SHE reference. Converting to RHE and using a value of 1.23 eV per transferred electron would be easier to understand.

Response: We thank the reviewer for this careful examination and expert comment.

We strongly agree with the reviewer's suggestion, and we have consistently converted all values and equations to the RHE scale, which includes revisions in both the manuscript (Pages 9-11) and SI (Pages S4-S6).

3. How were the free energies of molecular species (H_2O , H_2 , O_2) obtained, as given in Table S2?

Response: We thank the reviewer for raising this important question.

The free energy of H_2 (G_{H_2}) was calculated as an isolated gas molecule (in a large unit box of $12 \times 12 \times 15 \text{ \AA}^3$) with the free energy contributions from translation, rotation, and vibration assuming the ideal-gas approximation. The free energy of $H_2O(l)$ ($G_{H_2O(l)}$) was calculated in the same way as G_{H_2} but corrected with a $k_B T \ln p/p^\circ$ term that corresponds to the saturation vapor pressure ($p/p^\circ = 0.031$) in equilibrium with liquid water at room temperature. Finally, the free energy of O_2 (G_{O_2}) was derived from the expression of $G_{O_2} = 4.92 \text{ eV} + 2G_{H_2O(l)} - 2G_{H_2}$, where 4.92 eV is the experimental value

for the OER reaction free energy. To clarify this, we have revised the SI (Page S4) accordingly.

4. Grand-canonical DFT method: What is the value of the Debye length and other relevant VASPsol parameters? The authors assume a linearized relationship between the grand potential Ω and the electrode potential U . However, this neglects capacitive effects, since the interfacial capacitance defines the curvature of the $\Omega(U)$ curve. Is it justified to neglect the capacitive effect? Finally, how was the GC-DFT method combined with NEB to obtain the transition state under constant potential conditions?

Response: We thank the reviewer for this constructive and expert comment.

Regarding the VASPsol parameters, we used a Debye length of 3.0 Å that corresponds to a 1 M ionic strength and a dielectric constant of 78.4 for the aqueous solution. We have added these details in the manuscript (Page 21) accordingly.

Regarding the assumed linear relationship between Ω and U , we justified it as follows. We performed both linear and quadratic fitting of $\Delta\Omega(U)$ for all the reaction energies in AEM on NiOOH, as shown in **Figure R6** below, and the two types of fitting render essentially overlapping curves. **Table R3** below lists the fitted parameters, and the small fitted coefficients for the quadratic term imply that the capacitive effect is negligible here. In addition, **Table R3** lists the differences between $\Delta\Omega$ values at $U = 1$ V calculated from the two types of fitting, all of which are small (~ 0.01 eV). We have added **Figure R6** as Figure S15 and **Table R3** as Table S6 in the SI and revised corresponding discussion in the SI (Page S6) to point the readers to the justification.

Regarding the transition state (TS) search, we first located the TS structures using the NEB and dimer methods within the implicit electrolyte model, and then performed the single-point GC-DFT calculations upon these structures. Thus, we approximated the constant-potential TS structures for the chemical O-O coupling steps with the constant-charge counterparts to calculate the explicit dependences of their free energies on the applied potential. To clarify this, we have revised the manuscript (Page 21) accordingly.

Figure R6. Linear and quadratic fitting of $\Delta\Omega(U)$ for all the reaction energies in AEM on NiOOH. The two types of fitting render essentially overlapping curves, and Table R2 lists the fitted parameters.

Table R3. Fitted parameters for the linear fitting with the formula $\Delta\Omega = b_0 + b_1U$ and the quadratic fitting with the formula $\Delta\Omega = b_0 + b_1U + b_2U^2$. The small b_2 values imply that the capacitive effect is negligible here. The last column lists the differences between $\Delta\Omega$ values (in eV) at $U = 1$ V calculated from the two types of fitting, all of which are small.

Steps	Linear Fitting		Quadratic Fitting			Difference at $U = 1$ V
	b_0	b_1	b_0	b_1	b_2	
S0 to S1	0.282	0.003	0.282	0.009	-0.003	0.002
S1 to S2	0.831	0.100	0.837	0.064	0.018	-0.012
S2 to S4	-0.130	-0.198	-0.124	-0.208	0.005	0.001
S4 to S0	-0.984	0.095	-0.995	0.136	-0.020	0.009

5. *Microkinetic modeling: I am concerned about the pre-exponential factor used in the expression for the rate constants at line 380. The authors simply use the typical (kT/h) factor. However, additional pre-factors should be needed to account for the correct normalization of the rate constants with respect to the reference state used for DeltaOmega. Particularly, the rate equations at line 365 explicitly include only the coverages of surface intermediates, but the reaction steps also involve non-adsorbed reactants, namely H₂O, OH⁻ etc. Apparently, the respective concentrations of non-adsorbed species are lumped into the rate constants, which, in principle, is ok if these concentrations are constant. However, this must affect the respective pre-exponential factors, so I'm surprised about the simple (kT/h) used by the authors. In my opinion, the expression for the rate constants must be fundamentally motivated and explicitly related to the form of the rate equations at line 365 and the CHE reference for the DFT-computed DeltaOmega.*

Response: We thank the reviewer for this inspiring and expert comment.

First, the ($k_B T/h$) factor from the transition state theory (TST) has been shown to render quantitative predictions for heterogeneous catalysis by previous studies such as the classic work by Nørskov *et al.* [*Science* **307**, 555-558 (2005)]. Recently, Melander *et al.* proved that this ($k_B T/h$) factor also holds for the reactions at the electrochemical interfaces (open systems), which is the grand-canonical ensemble TST (GCE-TST) [*J. Electrochem. Soc.* **167**, 116518 (2020)].

Second, the intriguing question raised by the reviewer about the effects of concentrations of non-adsorbed reactants (H₂O and OH⁻) is related to how we write the expression of rate constant.

If using the form in our manuscript as follows,

$$k_{ij} = k_B T/h \cdot \exp(-\Delta\Omega^\ddagger/k_B T),$$

the concentrations of non-adsorbed reactants, which determines their chemical potentials (free energies), are explicitly included in $\Delta\Omega^\ddagger$, so the pre-exponential factor is simply $k_B T/h$.

Because the contributions from the concentrations to the free energy are entropic terms, we can write the above expression into a different form. Take [OH⁻] as an example (assuming OH⁻ is a reactant), its contribution to the free energy barrier can be singled out as,

$$\Delta\Omega^\ddagger = \Delta\Omega^{\ddagger*} - k_B T \ln[\text{OH}^-],$$

where $\Delta\Omega^{\ddagger*}$ is simply the free energy barrier without including the contribution from [OH⁻].

Consequently, the expression of the rate constant can be written as,

$$k_{ij} = k_{\text{B}}T/h \cdot [\text{OH}^-] \cdot \exp(-\Delta\Omega^{\ddagger}/k_{\text{B}}T),$$

and the pre-exponential factor now becomes $k_{\text{B}}T/h \cdot [\text{OH}^-]$, which explicitly includes the concentration of OH^- .

The two different forms are equivalent, and we used the first one with the simple $k_{\text{B}}T/h$ pre-exponential factor in this work.

Nevertheless, we strongly agree with the reviewer that the pre-exponential factor in electrochemistry can be complicated by various factors at the electrochemical interfaces, as reviewed in *Angew. Chem. Int. Ed.* **57**, 7948-7956 (2018), which thus needs further theoretical developments.

To clarify this, we have added the above discussion into SI as a Supplementary Note (Page S8) and revised the manuscript (Page 22) to point the readers to it in SI.

6. Electrochemical steps: Figure S9: The authors appear to assume that the barrier height changes 1:1 with the free-energy change between initial and final state. Effectively, this corresponds to a symmetry factor of $\beta = 1$. Typically, symmetry factors of electrochemical steps are rather around 1/2. How is the assumption of $\beta = 1$ justified?

Response: We thank the reviewer for this careful examination and expert comment.

In Figure S9, we aimed to justify the approximation we used for the barriers of electrochemical steps featuring proton transfer by testing a rigid minimal barrier ($\Delta\Omega_{\text{min}}^{\ddagger}$) imposed on all the barriers ($\Delta\Omega^{\ddagger}$) of electrochemical steps, which is defined as follows,

$$\Delta\Omega^{\ddagger} = \begin{cases} \Delta\Omega_{\text{min}}^{\ddagger}, & \text{for } \Delta\Omega \leq 0 \\ \Delta\Omega_{\text{min}}^{\ddagger} + \beta \cdot \Delta\Omega & \text{for } \Delta\Omega > 0 \end{cases}$$

where $\Delta\Omega$ is the reaction free energy of the electrochemical step. In order to test the most extreme scenario, we chose $\beta = 1$ (as pointed out by the reviewer), because this gives the fastest increase in $\Delta\Omega^{\ddagger}$ as $\Delta\Omega$ increases (when $\Delta\Omega > 0$). We did not intend to justify $\beta = 1$, but only wanted to test the most extreme scenario. Our test shows that $\Delta\Omega_{\text{min}}^{\ddagger}$ has little influence on the kinetics for $\Delta\Omega_{\text{min}}^{\ddagger}$ increasing from 0 to 0.40 eV (Figure S9b).

But we strongly agree with the reviewer that typically $\beta \approx 0.5$, so we have also performed a new test with $\beta = 0.5$ and added Figure S16 (shown below as **Figure R7**) for it. This new test also shows that $\Delta\Omega_{\text{min}}^{\ddagger}$ has little influence on the kinetics for $\Delta\Omega_{\text{min}}^{\ddagger}$ increasing from 0 to 0.50 eV (**Figure R7b**).

To clarify this, we have revised the caption of Figure S9 (Pages S21-22) and added **Figure R7** as Figure S16 to the SI.

Figure R7. (a) Illustration of the rigid minimal barrier ($\Delta\Omega_{\min}^{\ddagger}$) of 0.40 eV we impose on the barrier of electrochemical step with $\Delta\Omega = 0$ eV at $U = 0.8$ V. The barrier ($\Delta\Omega^{\ddagger}$) of electrochemical step is related to $\Delta\Omega_{\min}^{\ddagger}$ and the reaction free energy ($\Delta\Omega$) via the expression below,

$$\Delta\Omega^{\ddagger} = \begin{cases} \Delta\Omega_{\min}^{\ddagger}, & \text{for } \Delta\Omega \leq 0 \\ \Delta\Omega_{\min}^{\ddagger} + \beta \cdot \Delta\Omega & \text{for } \Delta\Omega > 0 \end{cases}$$

So, $\Delta\Omega_{\min}^\ddagger$ sets the lower bound for the barrier when $\Delta\Omega \leq 0$ and becomes an additional rigid term to the barrier when $\Delta\Omega > 0$. Here we test the typical $\beta = 0.5$. (b) Current density as the function of overpotential η_0 with different rigid minimal barrier ($\Delta\Omega_{\min}^\ddagger$) imposed on the electrochemical steps in OER on NiOOH. Note that the current density and thus the catalytic performance has little dependence on the $\Delta\Omega_{\min}^\ddagger$ of electrochemical step up to 0.5 eV.

7. AEM: The O-O coupling step is considered with H₂O as reactant. Alternatively, OH⁻ could serve as a reactant for this step in alkaline conditions. How would the results be affected by this choice? Is the microkinetic model able to discern these two alternatives and predict the dominant one?

Response: We thank the reviewer for this inspiring comment.

This led us to carefully examine the transition states (TSs) of O-O coupling in AEM, and we found that the reactant H₂O in all these TSs is already deprotonated by the bridging O site before reaching the TS, and the imaginary vibrational mode in each TS is essentially the stretching mode between the two O atoms that are forming the new O-O bond (we have added a new Figure S17 to show this more clearly). This indicates that the O-O coupling started with OH⁻ (the OH⁻ pathway) might share the same TS as that started with H₂O (the H₂O pathway), although the OH⁻ pathway should have an additional intermediate state with the bridging O site protonated. However, we argue that it is less likely for OH⁻ to serve as the reactant even at pH = 14, because the concentration of OH⁻ (1 M) is overwhelmed by that of H₂O (56 M), which is likely to result in a worse kinetics for the OH⁻ pathway assuming the same TS for the two pathways. If we included both pathways, the microkinetic modeling should be able to discern the two and predict the dominant one, which is likely to be the H₂O pathway. We have included the above discussion in the caption of the new Figure S17 in the SI (Page S31-32) for the readers' benefit.

8. Comparison with experiment (Table 1): For each material, the authors use the experimental overpotential from only one respective study for comparison with their modeling results. There are many experimental reports on NiOOH-based OER catalysts, so a more extensive literature comparison would be welcome. Regarding Tafel slopes, given the scattering of experimentally reported values, are they really useful for validating the model results?

Response: We thank the reviewer for this constructive comment.

Regarding the overpotentials, we wanted to use the experimental data of fair quality, and we had a few considerations. For the pure NiOOH, there is a well-known problem in the experiments, which is the Fe contamination [*J. Am. Chem. Soc.* **136**, 6744-6753 (2014)], so we chose the overpotential value from the classic work [*J. Am. Chem. Soc.* **137**, 1305-1313 (2015)] that was based on Fe-free NiOOH. For Ni(Co)OOH and Ni(Fe)OOH, we focused on the data that are of high quality and also are from the same work for fair comparison, which led us to the classic work [*Nat. Commun.* **5**, 4477 (2014)]. In addition, we resorted to excellent reviews [*Chem. Soc. Rev.* **46**, 337-365 (2017); *Adv. Energy Mater.* **6**, 1600621 (2016)], but with the considerations above, we did not manage to find more suitable experimental data. Thus, we just listed these values for the overpotentials in Table 1, and a more exhaustive literature search of experimental works is really beyond the scope of this study.

Regarding the Tafel slopes, the experimental works we used for the overpotentials did not report the Tafel slopes, so we had to loosen up our considerations, which led us to find a variety of values for the Tafel slopes as shown in Table 1. We agree with the reviewer that the Tafel slopes vary widely in the experimental reports, which may arise from various complications including the specific adsorptions of particular ions [*Energy Environ. Sci.* **9**, 1734-1743 (2016); *Angew. Chem. Int. Ed.* **56**, 8652-8656 (2017)], and this may compromise the comparison for validating our methodology. To emphasize this, we have revised the manuscript (Page 17) accordingly.

9&10. Several figures (in manuscript and SI) are difficult to read due to small size and poor resolution.

The resolution and readability should be improved.

Lines 165-167: This sentence is difficult to understand, please rephrase. What do the authors mean with the "PLS is not turned on"?

Response: We thank the reviewer for the constructive suggestions.

We have changed Figures 2-5 and Figures S3, S9 into with higher resolutions or larger font sizes, and we have rephrased the sentence on Page 11 to "PLS remains endothermic ...".

REVIEWERS' COMMENTS

Reviewer #1 (Remarks to the Author):

In their revised manuscript, Wang et al. have made considerable effort to better outline related work, clarify some issues raised by the referees, add additional data to make their arguments more conclusive and to sharpen their focus in explaining the novelty of their work.

I find it especially helpful that the authors have chosen to add clarifying statement and details for almost every point raised by the referees, so that the critical reader can assess the details of the work much better than in the original version.

Hence, I would consider this work publishable in Nature Communications given that the other referees agree as well.

Reviewer #2 (Remarks to the Author):

The authors have successfully addressed all the issues I raised. I, therefore, recommend it for publication.

Reviewer #3 (Remarks to the Author):

The authors of the manuscript "Potential-dependent Transition of Reaction Mechanisms for Oxygen Evolution on Layered Double Hydroxides" carefully responded to my previous comments and addressed all relevant points in the revised manuscript. I consider the microkinetic approach for coupled reaction networks with potential-dependent kinetics an important step towards a more realistic description of electrocatalytic processes. I recommend the present manuscript for publication in Nature Communications.

Point-by-point Response to the Reviewers' Comments

Reviewer #1

In their revised manuscript, Wang et al. have made considerable effort to better outline related work, clarify some issues raised by the referees, add additional data to make their arguments more conclusive and to sharpen their focus in explaining the novelty of their work.

I find it especially helpful that the authors have chosen to add clarifying statement and details for almost every point raised by the referees, so that the critical reader can assess the details of the work much better than in the original version.

Hence, I would consider this work publishable in Nature Communications given that the other referees agree as well.

Response: We greatly appreciate the reviewer's approval of our manuscript for publication, and we would like to express our sincere gratitude to the reviewer for the time and efforts dedicated to thoroughly reviewing our work and providing constructive and expert comments that have contributed to greatly improving the quality of this manuscript.

Reviewer #2

The authors have successfully addressed all the issues I raised. I, therefore, recommend it for publication.

Response: We greatly appreciate the reviewer's approval of our manuscript for publication, and we would like to express our sincere gratitude to the reviewer for the time and efforts dedicated to thoroughly reviewing our work and providing constructive and expert comments that have contributed to greatly improving the quality of this manuscript.

Reviewer #3

The authors of the manuscript "Potential-dependent Transition of Reaction Mechanisms for Oxygen Evolution on Layered Double Hydroxides" carefully responded to my previous comments and addressed all relevant points in the revised manuscript. I consider the microkinetic approach for coupled reaction networks with potential-dependent kinetics an important step towards a more realistic description of electrocatalytic processes. I recommend the present manuscript for publication in Nature Communications.

Response: We greatly appreciate the reviewer's approval of our manuscript for publication, and we would like to express our sincere gratitude to the reviewer for the time and efforts dedicated to thoroughly reviewing our work and providing constructive and expert comments that have contributed to greatly improving the quality of this manuscript.